# Performance Improvement and Microstructure Characterization of Cement-Stabilized Roadbase Materials Containing Phosphogypsum/Recycled Concrete Aggregate

**DOI:** 10.3390/ma16196607

**Published:** 2023-10-09

**Authors:** Yang Wu, Xiaoya Bian, Jie Liu, Ruan Chi, Xuyong Chen

**Affiliations:** 1School of Civil Engineering and Architecture, Wuhan Institute of Technology, Wuhan 430073, China; 22104010088@stu.wit.edu.cn (Y.W.); bianxy@wit.edu.cn (X.B.); werewolves@163.com (J.L.); 2Hubei Provincial Engineering Research Center for Green Civil Engineering Materials and Structures, Wuhan Institute of Technology, Wuhan 430073, China; 3Hubei Three Gorges Laboratory, Yichang 443000, China; rac@wit.edu.cn

**Keywords:** roadbase, phosphogypsum, recycled concrete aggregate, sodium metasilicate nonahydrate, performance evaluation, microstructure characterization

## Abstract

The proper reutilization of the phosphogypsum (PG) by-product derived from the production of phosphoric acid and recycled concrete aggregate (RCA) from waste concrete in roadbase materials is of great necessity and importance. This investigation tried seeking a new approach to reuse them to high quality, including turning PG into calcinated PG (CPG) via washing and calcination, as well as adopting sodium metasilicate nonahydrate (SMN) to strengthen the roadbase materials of cement-stabilized CPG and RCA. Upon the mix design, with a series of experiments including unconfined compressive strength, the wet–dry cycle, freeze–thaw cycle, and scanning electron microscopy, the comprehensive effects of PG treatment, the CPG to RCA mix ratio, SMN dosage, wet–dry cycle and freeze–thaw cycle on the road performance of roadbase materials were well evaluated, and the traffic bearing capacity and microstructure characteristics were also analyzed. The results demonstrate that the 7 d unconfined compressive strength of CPG/RCA roadbase materials can reach 5.34 MPa as the CPG and SMN dosage are 20% and 11%, respectively, which meets the requirements of an extremely and very heavy traffic grade. After five wet–dry cycles and freeze–thaw cycles, the resistance of the CPG/RCA roadbase materials to moisture and frost was significantly improved as 11% SMN was added. Meanwhile, SMN contributes to the reduction in crack width and densifies the microstructure of CPG/RCA roadbase materials. The research results can be used to provide new guidance for building more durable roadbase materials.

## 1. Introduction

Phosphogypsum (PG) is mainly the calcium sulfate dihydrate (CaSO_4_⋅2H_2_O), which is formed as a by-product of the production of fertilizer, particularly phosphoric acid [1,2,3]. According to current statistics, the global stockpile of PG has reached 6 billion tons, with an increasing rate of 200 million tons per year [4]. Because of this, large amounts of PG are disposed of at production sites for storage. As of now, reports are stating that they are recycled and reused in various fields such as chemistry, agriculture, building, etc., though the consumption is still considered very limited [5]. In addition to PG, waste concrete is also one of the most important municipal solid wastes that are recycled from buildings, roads, and bridges [6,7,8]. Similarly, its disposal methods are still landfilling and stockpiling, which, as of now, has caused the large occupation of land resources and the pollution of the ecological environment [9,10]. Therefore, it is of great importance and urgency to find a new pathway to collectively recycle PG and waste concrete on a large scale, for example, in the application of roadbase materials [11,12,13].

As PG is very susceptible to moisture and unfriendly to the environment, their application is actually difficult in different areas [14,15]. Generally, harmless treatment should be considered at first by a series of processes, including physical, chemical and calcination methods, and afterward, some hemihydrate PG (CaSO_4_⋅0.5H_2_O) can be obtained [16,17,18]. Regarding its application in roadbase materials, researchers worldwide have carried out a series of studies. For instance, Mohammad et al. [19] found that the partial replacement of Portland cement using PG can still reach an applicable strength requirement in rigid pavements. Ding et al. [20] concluded that an appropriate amount of PG can be used in the roadbase mix for application, but excessive PG causes significant damage, especially moisture-induced damage, to the base. Zhang et al. [21] found that adding an appropriate amount of PG (6%) can effectively improve the performance of lime–fly ash-crushed stone roadbase materials, proving that it is feasible to use PG instead of lix–fly ash to stabilize gravel and lime in the gravel mixture. The mentioned studies indicate that there is huge potential to apply PG to roadbase materials.

As for the research on RCA in roadbase materials [22,23], using RCA as an inorganic mixture is an effective way to recycle waste concrete resources at present. Kox et al. [24] applied RCA to concrete pavement, and the results showed that the replacement rate of coarse aggregate at 40% would not have adverse effects. It has little effect on concrete aggregate durability and freeze–thaw resistance. Poon et al. [25] used recycled concrete aggregate and crushed clay brick as the base mixture, and 100% recycled aggregate significantly increased the optimal water content and maximum dry density of the mixture. However, by adding crushed clay bricks instead of recycled aggregate, the index of the mixture could be reduced, and the immersed California Bearing Ratio (CBR) value of the mixture was greater than 30%, meeting the requirements of the specification. Under sandy soil, columns of different lengths were constructed using the compacted RCA structure. Soil pH values at different depths were measured in the experiment. The results show that the leachate of basic RCA can be fully buffered after carbonization. It has been proved that alkaline RCA leachate does not cause corrosion to metal-clad steel culverts in underground soil [26]. However, there are few reports on the high-performance utilization of PG and RCA in roadbase materials at the same time, and further in-depth research is needed.

To better recycle PG and RCA into roadbase materials, this study aims to provide a new pathway to realize their application in high performance. The designed process includes the pretreatment of PG by washing and calcination to prepare calcinated PG (CPG) and the use of sodium metasilicate nonahydrate (SMN) to enhance roadbase materials containing CPG and RCA. Through mixed design and test characterizations such as compressive strength, the wet–dry cycle, freeze–thaw cycle, and scanning electron microscope, the effects of the PG treatment mode, SMN dosage, wet–dry cycle, and freeze–thaw cycle on the performance of roadbase materials are studied, and the traffic bearing capacity and microstructure characteristics of roadbase materials are also analyzed.

## 2. Materials and Methods

### 2.1. Materials

PG was supplied from a local factory in Hubei province, China, and presented as a moisture-induced agglomerated powdery appearance and a dry fine powdery outlook after calcination (see Figure 1). After testing, its chemical composition is shown in Table 1. For calcinated PG (CPG), the particle size distribution is presented in Table 2.

The used coarse aggregates were RCA with a particle size of 4.75–19.00 mm, which were composed of natural aggregates and adhered mortar, as shown in Figure 2. The used fine aggregate was natural river sand, which belonged to the middle sand with a particle size of 0~4.75 mm, recorded as NFA. The coarse and fine aggregates were provided locally. According to JGJ 52-2006 [27], the test results of technical indicators relating to coarse and fine aggregates are shown in Table 3.

Ordinary Portland cement is a locally supplied P.O. 42.5 grade product, and the performance results are tested and shown in Table 4, according to GB 175-1999 [28]. Sodium metasilicate nonahydrate (SMN) was purchased from Tianjin Sinopharm Group Chemical Reagent Co., Ltd., China. with the molecular formula Na_2_SiO_3_·9H_2_O, pH > 7; it is white, powdery and easily soluble in water.

### 2.2. Mix Design and Preparation Method

#### 2.2.1. Mix Design

First of all, PG was treated via washing and calcination (calcination temperature of 150 °C and 1 h) to obtain CPG. The graded aggregates, including RCA and NFA, were replaced by CPG at 10%, 20%, 30%, 40%, and 50% by weight, respectively, and the mix ratios of CPG/RCA roadbase materials were designed. At the same time, the maximum dry density (MDD) and optimal moisture content (OMC) were calculated by the test results of six samples for each group according to JTG E51-2009 [29], as shown in Table 5. Subsequently, according to the compressive strength test, after determining the optimal ratio of CPG to RCA, 3%, 5%, 7%, 9%, 11%, and 13% SMNs (by weight of CPG) were mixed into the mixture for different specimens. Finally, the optimal amount of SMN was determined based on the compressive strength test.

In order to further clarify the performance advantages of the designed CPG/RCA roadbase material, this study considered a comparative study of 0CPG-57RCA, PG-RCA, and xCPG-yRCA-zSMN. Among them, x, y, and z produce the best mix ratio, as determined by the compressive strength test.

#### 2.2.2. Preparation Method

According to JTG E51-2009, standardized Φ100 mm × 100 mm samples were first prepared and then put into the curing device with a constant temperature of 20 ± 2 °C and relative humidity of 95% for 7 d and 28 d. After curing, the samples were removed and soaked in water at 20 ± 2 °C for 24 h. After the removal of surface water, unconfined compressive strength tests were carried out.

### 2.3. Testing and Characterization

#### 2.3.1. Unconfined Compressive Strength

According to JTG E51-2009, the Φ100 mm × 100 mm cylindrical samples were molded by static pressure and cured for 7 d, and six samples in each group were prepared for the test. After curing, the unconfined compressive strength for each sample was tested with a loading rate of 1 mm/min (see Figure 3). Upon testing, the influence of the PG/RCA mixing ratio and SMN on the confined compressive strength of PG/RCA roadbase materials was evaluated. Unconfined compressive strength can be calculated as follows:$$R_{c}=\frac{P}{A}$$
where $R_{c}$ is the unconfined compressive strength (MPa); P is the maximum pressure (N) of the sample failure; and A is the cross-sectional area of the sample (mm^2^).

According to JTGT F20-2015 [30], the traffic-carrying grade of PG/RCA roadbase materials was evaluated and divided based on the 7 d unconfined compressive strength result, as shown in Table 6.

#### 2.3.2. Wet-Dry Cycle Test

The specimens were first cured for 7 d under curing conditions and then placed in water for 1 d and dried in an oven at 40 °C for 1 d. The whole process is regarded as a complete wetting and drying cycle. Six samples were prepared for the test in each group. Furthermore, the effects of the PG treatment mode, the ratio of PG to RCA, the amount of SMN dosage, and wet–dry cycles on the compressive strength of the target roadbase materials were studied to analyze the water stability of PG/RCA base materials with moisture-changing conditions.

#### 2.3.3. Freeze–Thaw Cycle Test

According to JTG E51-2009, six samples were prepared in each group, and after curing for 28 d, the cured samples were individually put into the −18 °C cryogenic box for a 16 h freezing test. After this, the samples were taken out and immediately placed into a 20 °C sink for melting for 8 h. After melting, the freeze–thaw cycle continued five times. Based on this, the effects of the PG processing mode, PG-RCA ratio, SMN dosage, and freeze–thaw cycle times on the unconfined compressive strength of the target base material were studied to evaluate the frost resistance of PG/RCA base materials.

#### 2.3.4. Scanning Electron Microscope Test

Before the test, 28 d-cured samples were cut into rectangular sizes no larger than 5 mm and then went through a gold-plated process. During the test, the following conditions were set at a high vacuum mode, a physical working distance of ~10 mm, and an accelerated voltage of 20.00 kV [31]. The final images were captured at 5 kx for comparative studies to evaluate if the designed method could help improve the microstructures of roadbase materials containing CPG and RCA.

### 2.4. Research Flowchart

To clearly understand this study, a research flowchart was formed and is displayed in Figure 4. It includes the pretreatment of PG, the preparation of roadbase materials containing CPG and RCA, and the test methodologies for the final assessment of its traffic carrying capacity. In addition, the summary of name abbreviations is presented in Table 7.

## 3. Results and Discussion

### 3.1. Unconfined Compressive Strength Results

#### 3.1.1. Effect of Different CPG Contents

Figure 5 shows the influence of CPG contents on the 7 d and 28 d unconfined compressive strength of CPG/RCA roadbase materials. As displayed, with the increase in the CPG dosage, the 7 d and 28 d unconfined compressive strength of CPG/RCA roadbase materials gradually decreased, and when the CPG content reached 50%, it could still meet the medium and light traffic carrying level (the minimum strength threshold: 3 MPa). In addition, when the CPG content exceeded 20%, the 7 d unconfined compressive strength could hardly meet the heavy-traffic carrying level (the minimum strength threshold: 4 MPa). These results show that, regardless of the curing period, the increased content of CPG gradually reduces the unconfined compressive strength of CPG/RCA roadbase material. This is because the excessive use of CPG reduces the proportion of coarse aggregates, leading to roadbase strength reduction due to the roadbase structure transformation from skeleton cavitation to skeleton suspension. Therefore, this study took 20CPG-45RCA as the basic research object for further discussion and analysis to guarantee the heavy traffic level.

#### 3.1.2. Effect of Different SMN Contents

Figure 6 presents the effect of SMN contents on the 7 d unconfined compressive strength of 20CPG-45RCA roadbase material. It is clear that, as the SMN content increased, the 7 d unconfined compressive strength of 20CPG-45RCA gradually increased, and as its content reached 9%, the compressive strength of 20CPG-45RCA-9SMN reached 5.03 MPa, meeting the extreme and extra-duty traffic carrying level (minimum strength threshold: 5 MPa). When its content increased to 11%, the compressive strength of 20CPG-45RCA-11SMN rose to 5.37 MPa, and then with 13% SMN, the compressive strength of 20CPG-45RCA-13SMN did not increase significantly, reaching only 5.41 MPa. These results indicate that the increasing SMN content gradually increased the compressive strength of 20CPG-45RCA, but excessive SMN (more than 11%) had a very small contribution to the strength improvement. This is because the chemical reactions between SMN and CPG promote reaction products of calcium silicate hydrates (CSH), calcium aluminate hydrate crystals, and silicate gels, which directly leads to the enhanced bonding property of recycled aggregates, whereas the excessive use of SMN also leads to the volume expansion of products such as alumite in the inner pores of roadbase material, easily causing stress concentration toward damage. Therefore, this study selected the SMN content of 11% as the best choice for further study.

#### 3.1.3. Compressive Strength Comparison of Different PG/RCA Roadbase Materials

Figure 7 reflects the 7 d and 28 d unconfined compressive strength of different PG/RCA roadbase materials. As displayed and compared to 0CPG-57RCA, the compressive strengths of 20PG-45RCA can be significantly reduced. However, compared to 20PG-45RCA, the compressive strength of 20CPG-45RCA increases to a certain extent, but it is still less than the 0CPG-57RCA level. As 11% SMN is added, the 7 d and 28 d compressive strengths of 20CPG-45RCA are significantly improved, from 4.08 MPa to 5.37 MPa and from 4.92 MPa to 6.65 MPa, respectively, both of which are higher than the corresponding strength of 0CPG-57RCA (4.79 MPa and 6.23 MPa), reaching an extreme and extremely heavy traffic level. These results demonstrate that untreated PG has a significantly negative impact on the compressive strength of RCA roadbase material. As for the calcination of PG, such effects can only be reduced limitedly, while an appropriate amount of SMN can promote a remarkable increase in the compressive strength of PG/RCA roadbase material.

### 3.2. Water Stability

Figure 8 shows the influence of the number of dry and wet cycles on the unconfined compressive strength of PG/RCA base material after curing for 7 days. As can be seen from the figure, with the increase in the number of dry and wet cycles, the compressive strength of all materials presents a trend of first increasing and then decreasing. In the second cycle, the strength of all materials reaches the maximum value, among which 0CPG-57RCA and 20CPG-45RCA-11SMN can reach the grade of an extremely and very heavy traffic load. When this fifth cycle is carried out, the strength of 20PG-45RCA is 2.87 MPa and can no longer meet the requirements of medium and light traffic load class. The strengths of 20CPG-45RCA, 0CPG-57RCA, and 20CPG-45RCA-11SMN are 3.79 MPa, 4.60 MPa, and 5.38 MPa, respectively, which meet the load levels of medium and light traffic, heavy traffic, and extremely heavy traffic. These results show that the strength of the base material can be further improved after 7 days of maintenance at the early stage of the dry and wet cycle, but increasing the number of dry and wet cycles can lead to a decline in the strength of the material or its traffic load class. On the other hand, the addition of SMN provides better water stability for the mixture, which is similar to the findings by Shen et al. [32]. This is because the early phase of the wet and dry cycle can continue to promote the hydration reaction of the material, while the late phase of the wet and dry cycle is affected by the erosion of water, resulting in a decrease in its strength. In general, an appropriate SMN can significantly increase the compressive strength of CPG/RCA base material so that it has good resistance to water damage, and it can also maintain the level of an extremely heavy and extra-heavy traffic load.

### 3.3. Frost Resistance

Figure 9 exhibits the effect of freeze–thaw cycles on the unconfined compressive strength of PG/RCA roadbase material cured for 28 d. It was found that before the freeze–thaw cycle, the unconfined compressive strength of all base materials exceeded 4.00 MPa, meeting the carrying levels of heavy traffic and above. The unconfined compressive strength of 20PG-45RCA decreased from 4.03 MPa to 3.56 MPa after one freeze–thaw cycle, and the unconfined compressive strength of 20CPG-45RCA decreased from 4.92 MPa to 3.67 MPa after two freeze–thaw cycles, both of which could not meet the heavy traffic carrying level. 0CPG-57RCA and 20CPG-45RCA-11SMN can still reach 4.31 MPa and 4.65 MPa after five freeze–thaw cycles, respectively, which meets the heavy traffic carrying level. Moreover, 20CPG-45RCA-11SMN has higher compressive strength to meet a higher carrying level. After the freeze–thaw cycle, the proportion of micro-cracks and harmful pores in the base material increases. Although the addition of CPG can fill these pores, it reduces the cohesiveness of the cement paste. The curing effect of SMN can effectively reduce the generation of cracks, enhance the bonding force, and improve frost resistance, which is consistent with the results of Wang et al. [33]. These results indicate that an appropriate amount of SMN can significantly improve the frost resistance of 20CPG-45RCA, making it superior to the frost resistance of RCA roadbase material. The main reason for this is that the hydration reaction of SMN promotes the formation of a large amount of cementitious material, which plays a role in stabilizing CPG and RCA, further reducing the number of internal voids for lowered water erosion damage. These results are in compliance with the findings of Ruan et al. [34], combining bauxite tailings and sodium silicate to fully solidify the magnesium phosphate cement, which produces zeolite-like hydration product sodium silicate aluminate hydrate and makes the structure more compact while effectively reducing cracking.

Figure 10 shows the influence of freeze–thaw cycles on the residual compressive strength ratio of different PG/RCA roadbase materials. As displayed, with the increase in freeze–thaw cycles, the strength loss of 20PG-45RCA and 20CPG-45RCA increased rapidly. The residual compressive strength ratio after the third freeze–thaw cycle was 62.3% and 64.0%, respectively, which is lower than the requirements of the medium-freeze zone index (≥65%, JTG D50-2017) [35], indicating that the calcination of PG cannot improve the frost resistance of roadbase materials containing PG and RCA. Compared with 0CPG-57RCA, the residual compressive strength ratio of 20CPG-45RCA-11SMN could still reach 70.2% after the fifth freeze–thaw cycle, meeting the requirements of the heavy frozen zone index (≥70%). This indicates that the addition of SMN retains unconfined compressive strength at a higher level to samples after freeze–thaw cycles. The above results were similar to that provided by Saberian et al. [36]. To summarize, an appropriate amount of SMN can contribute to significantly improving the engineering properties of roadbase material while RCA and CPG were overall considered.

### 3.4. Microstructure Analysis

Figure 11 shows the microstructure of different PG/RCA base materials. As observed, some small cracks appeared in the microstructure of 0CPG-57RCA (Figure 11a), larger cracks appeared in 20PG-45RCA (Figure 11b), and larger cracks still existed in 20CPG-45RCA (Figure 11c), while the crack width of 20CPG-45RCA-11SMN was obviously smaller (Figure 11d). These phenomena identify that the addition of SMN can promote the hydration process to assist the stabilization of PG/RCA roadbase material by greatly improving the bonding properties between CPG and other components and reducing the development of cracks. These results are in compliance with the findings of Nasir et al. [37], who incorporated industrial waste-based pastes wherein adequate curing enabled the dissolution of precursor materials, which consequently led to the formation of a dense skeletal matrix, thereby yielding an enhancement in overall engineering. From these microstructural results, it has been proven that using SMN to help recycle amounts of CPG into roadbase materials is technically feasible and worthy of recommendation.

## 4. Conclusions

In this study, PG was first pretreated via washing and calcination to obtain CPG, and then SMN was used to enhance the treatment of cement-stabilized CPG and RCA roadbase materials. The effects of the PG treatment mode, SMN dosage, wet–dry cycle, and freeze–thaw cycle on the compressive strength of the roadbase materials were studied, and the traffic-bearing capacity and microstructure were also analyzed. The main conclusions are as follows:(1)The results of compressive strength show that pretreated PG can slightly reduce its adverse impact on the compressive strength of roadbase materials containing RCA. As 11% SMN is considered, its compressive strength can be significantly increased to 5.37 MPa, reaching the requirement of an extremely and very heavy traffic grade for application in the expressway and first-class highway.(2)The water stability results indicate that, compared to 0CPG-57RCA, 20CPG-45RCA-11SMN has better resistance to moisture, and its compressive strength still exceeds 5 MPa after five wet–dry cycles and meets the requirement of an extremely and very heavy traffic grade for application in the expressway and first-class highway.(3)The freeze–thaw test results state that compared to 0CPG-57RCA, 20CPG-45RCA-11SMN has better frost resistance. Its compressive strength and residual compressive strength ratio can still reach 4.65 MPa and 70.2% after five freeze–thaw cycles, respectively, meeting the requirements of a heavy traffic grade and heavy frozen area.(4)Microstructural analysis demonstrates that 11% SMN can effectively promote the occurrence of hydration reactions to reduce the crack width and pore generation rate of roadbase materials containing CPG and RCA with improved compactness.

## Figures and Tables

**Figure 1 materials-16-06607-f001:**
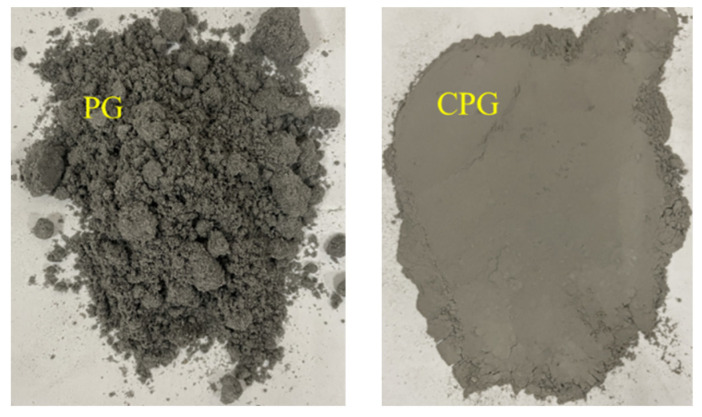
PG and CPG surface appearance.

**Figure 2 materials-16-06607-f002:**
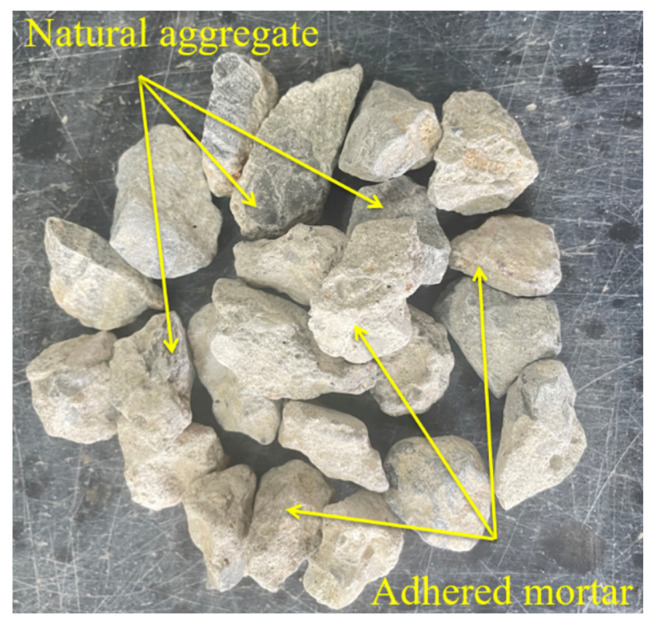
Surface appearance of RCA.

**Figure 3 materials-16-06607-f003:**
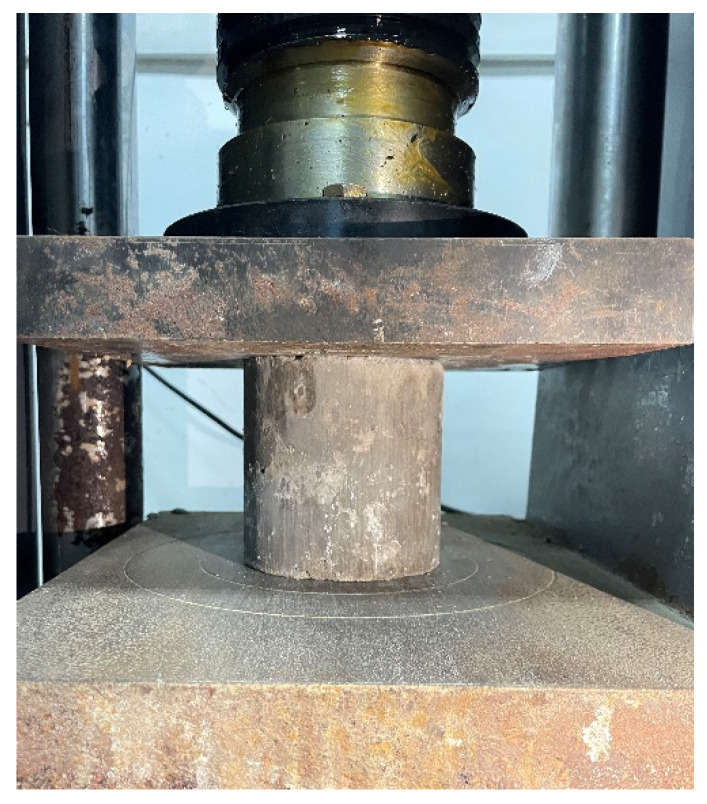
The unconfined compressive strength test of samples.

**Figure 4 materials-16-06607-f004:**
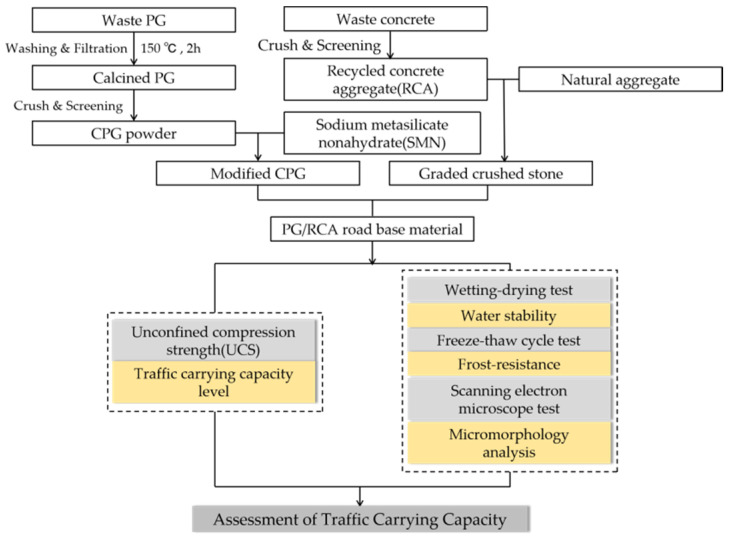
The research flow chart of this study.

**Figure 5 materials-16-06607-f005:**
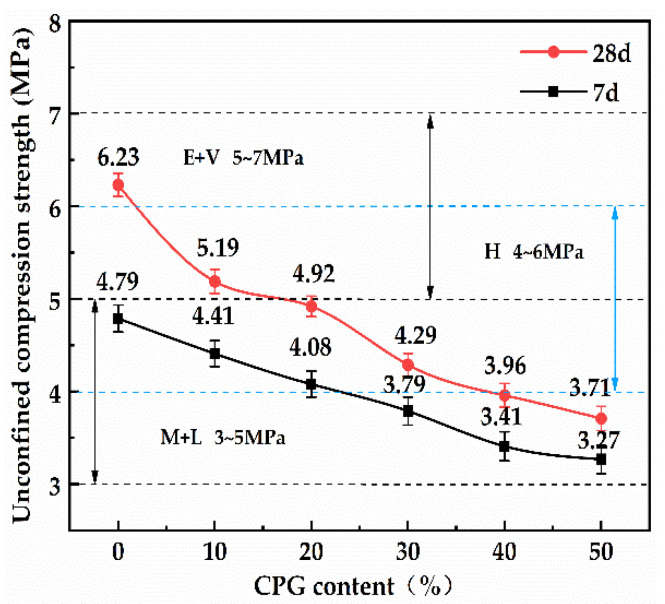
The 7 d and 28 d unconfined compressive strength of different CPG/RCA roadbase materials.

**Figure 6 materials-16-06607-f006:**
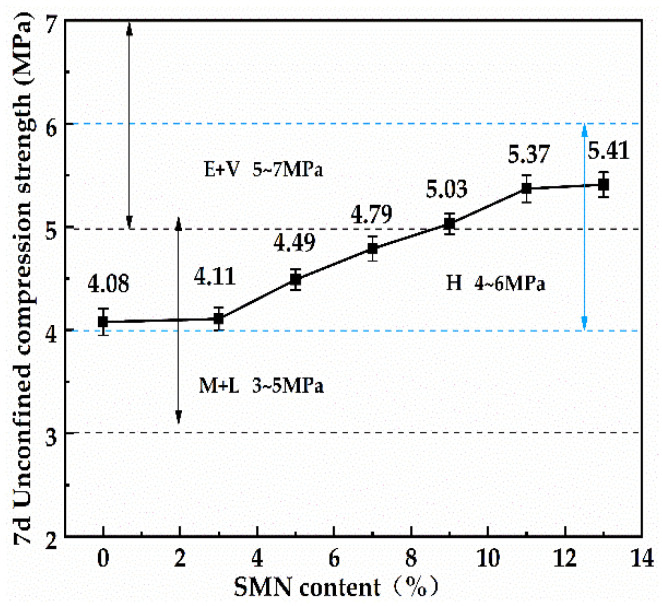
Effect of SMN contents on 7 d unconfined compressive strength for 20CPG-45RCA roadbase material.

**Figure 7 materials-16-06607-f007:**
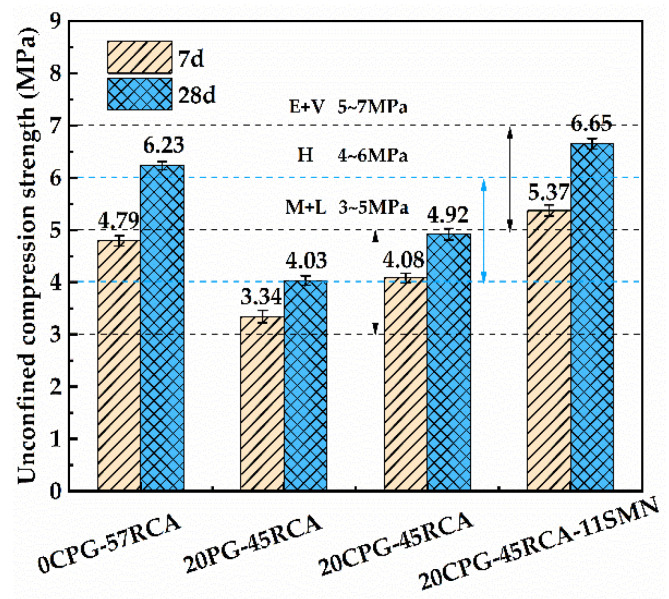
Unconfined compressive strength of different PG/RCA roadbase materials.

**Figure 8 materials-16-06607-f008:**
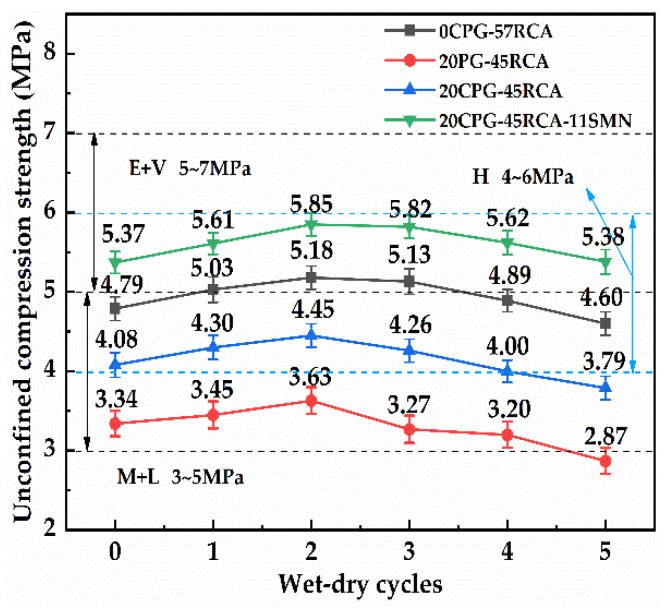
Effect of wet–dry cycle on the 7 d unconfined compressive strength of PG/RCA roadbase material.

**Figure 9 materials-16-06607-f009:**
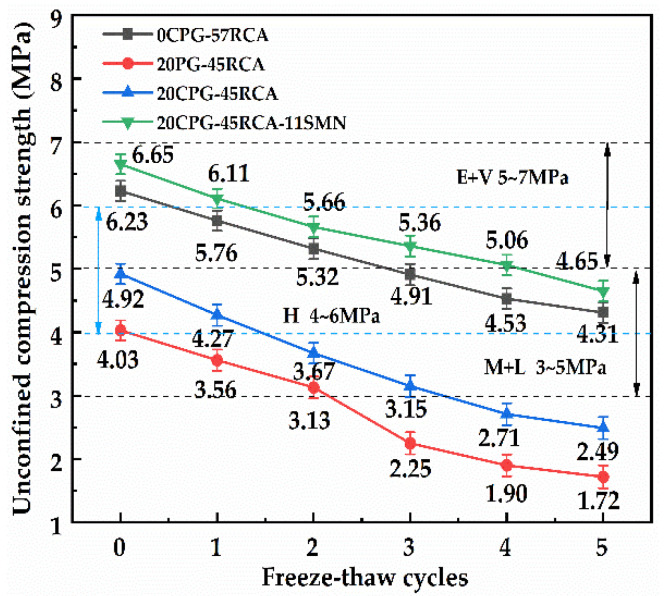
The effect of freeze–thaw cycles on the unconfined compressive strength of different PG/RCA roadbase materials cured for 28 d.

**Figure 10 materials-16-06607-f010:**
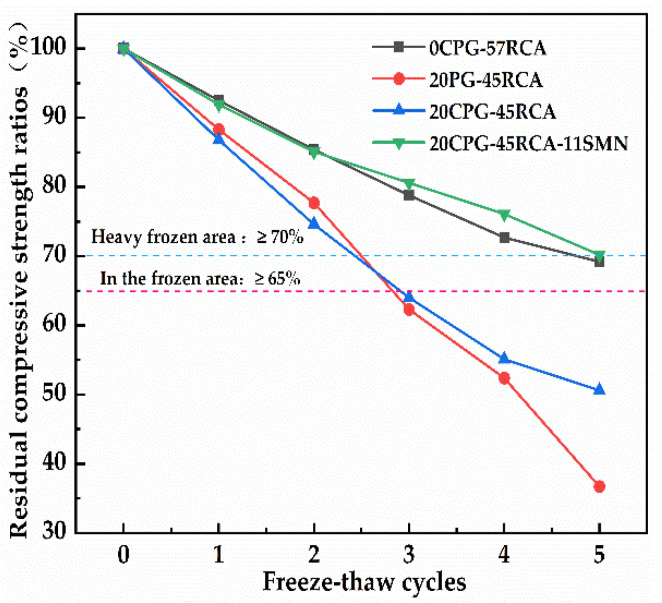
Effect of freeze–thaw cycles on residual compressive strength ratios of different PG/RCA roadbase materials.

**Figure 11 materials-16-06607-f011:**
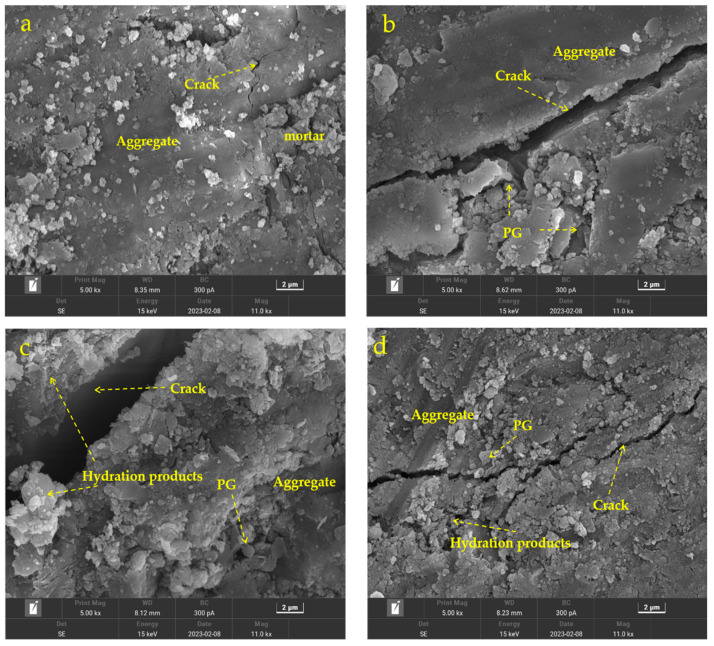
Microstructural images of different PG/RCA roadbase materials. (**a**) 0CPG-57RCA; (**b**) 20PG-45RCA; (**c**) 20CPG-45RCA; (**d**) 20CPG-45RCA-11SMN).

**Table 1 materials-16-06607-t001:** Chemical composition of PG.

Composition	CaO	SO_3_	SiO_2_	P_2_O_5_	Al_2_O_3_	MgO	As_2_O_5_	Cr_2_O_5_	BaO	F	Crystal Water
Percentage/%	30.72	40.20	4.63	1.51	2.25	0.01	0.59	0.01	0.04	0.15	18.29

**Table 2 materials-16-06607-t002:** Particle size distribution of CPG.

Particle Size/mm	2.36	1.18	0.6	0.3	0.15	0.075
Passing percentage/%	100	96.87	93.26	90.89	78.36	9.32

**Table 3 materials-16-06607-t003:** Physical performance indicators of aggregates.

Aggregates	Bulk Density (kg/m^3^)	Apparent Density (kg/m^3^)	Porosity (%)	Water Absorption Rate (%)	Crushing Value (%)	Mass Percent of Adhered Mortar to NA (%)	Fineness Modulus
RCA	1190	2560	0.47	6.1	15.9	25~30	—
NFA	1240	2597	—	—	—	—	2.94

**Table 4 materials-16-06607-t004:** Cement performance indicators.

Type	Standard Consistency Water Consumption (%)	Condensation Time (min)		Compressive Strength (MPa)	
		Initial Setting Time	Final Setting Time	3 d	28 d
P.O 42.5	25.9	215	279	24.5	48.2

**Table 5 materials-16-06607-t005:** Mix design of roadbase materials containing CPG/RCA.

Item	Cement (%)	CPG (%)	Aggregates (%)		OMC (%)	MDD (g/cm^3^)
			RCA	NFA		
0CPG-57RCA	5	0	57	38	6.73	2.272
10CPG-51RCA	5	10	51	34	5.97	2.059
20CPG-45RCA	5	20	45	30	6.17	2.044
30CPG-39RCA	5	30	39	26	6.65	1.999
40CPG-33RCA	5	40	33	22	7.50	1.931
50CPG-27RCA	5	50	27	18	8.44	1.869

**Table 6 materials-16-06607-t006:** Traffic carrying grade of roadbase material based with its 7 d unconfined compressive strength result (MPa).

Structural Layer	Highway Level	Extremely and Very Heavy Traffic (E + V)	Heavy Traffic (H)	Medium and Light Traffic (M + L)
Roadbase	Expressway and first-class highway	5.0~7.0	4.0~6.0	3.0~5.0
	Second-class and below highway	4.0~6.0	3.0~5.0	2.0~4.0

**Table 7 materials-16-06607-t007:** Summary of name abbreviations.

Name	Abbreviation
Phosphopypsum	PG
Calcination phosphopypsum	CPG
Sodium metasilicate nonahydrate	SMN
Rcycled concrete aggregate	RCA
Natural aggregate	NA
Natural coarse aggregate	NCA
Natural fine aggregate	NFA

## Data Availability

Data sharing is not applicable to this article.

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
