# Peer review of "Performance Improvement and Microstructure Characterization of Cement-Stabilized Roadbase Materials Containing Phosphogypsum/Recycled Concrete Aggregate"

_materials, 2023, doi:10.3390/ma16196607_

Round 1

Reviewer 1 Report

The comments of this reviewer are:

Line 54: “Regarding the comprehensive utilization of PG and RCA in roadbase materials [16,17]”, what conclusions did these references obtain? How the study presented differ from these references? What is the novelty respect to those references?

Line 75: “there are few reports of high-performance comprehensive utilization of roadbase materials carried out by PG and RCA at the same time” are those reports [16] and [17]? If not, include these reports and make and brief description and results obtained.

Line 85: “The research process of this study is shown in Fig. 1.” This is not the place to locate this figure. It should be in section 2.

Line 99: “which were composed of natural aggregates and adhered mortar” were not there any concrete particles?  if that the case and there were just natural aggregates and mortar it is not regular RCA.

Table 3: Bulk density instead of stack density. Please, which standards were used for Apparent density porosity… . Why there is no values for NFA porosity and water absorption?

Line 121: “(by weight of CPG)” if the substitution is based on the weight and the density differs, the volume varies. Have you considered this for justifying the results?

Table 5: The aggregates (RCA: NFA) are replaced by CPG when the particle size distribution is completely different, how much this affects to the properties studied?  The relation RCA: NFA=3:2 is in weight or in volume? Why did you consider that ratio?

Line 164: “standard sample of 28d” was it dried or at the conditions of curing?

In section 3 Results and discussion there is no single reference when it is stated, in the guide for authors in Materials journal, “Discussion: Authors should discuss the results and how they can be interpreted in perspective of previous studies and of the working hypotheses.” No adding any reference in this section diminishes the quality of the paper abruptly.

The subsection 3.4 Microstructure analysis could be located before with the purpose to justify the test results values.

The format of the reference list does not match with Materials requirement.

The quality of the English of the manuscript should be improved

Author Response

Reviewer #1:

Line 54: “Regarding the comprehensive utilization of PG and RCA in roadbase materials [16,17]”, what conclusions did these references obtain?  How the study presented differ from these references?  What is the novelty respect to those references?

Response: Thanks for your comment.

Reply to Question (1): These references mainly include: study on the application of hemihydrate PG (HH) produced by heat treatment in asphalt pavement mixture, and prove the feasibility of its application on pavement with consideration of environmental protection [16].

Columns of different lengths were constructed and containing compacted RCA under sandy soil. Soil pH values at different depths were tested during the test. The results showed that the leachate of alkaline RCA could be fully buffered by carbonation. It has been verified that alkaline RCA leachate will not cause corrosion of metal-coated steel culverts in underground soil [17].

Reply to Question (2): PG or RCA, as substitutes for pavement materials, should be used in highway construction worldwide. The feasibility of using PG and RCA as road base materials at the same time to study PG-RCA base materials will be considered. Based on these literatures, this study will study the application of the road base mixture prepared by CPG and RCA after PG calcined treatment in roads. In addition, the introduction has been revised, please check it in the article.

Reply to Question (3): These references study the application of PG or RCA in roadbase and pavement materials, and evaluate the strength and performance of PG and RCA as roadbase and pavement materials.

Line 75: “there are few reports of high-performance comprehensive utilization of roadbase materials carried out by PG and RCA at the same time” are those reports [16] and [17]?  If not, include these reports and make and brief description and results obtained.

Response: Thanks for your comments.

Reply to Question (1): These reports are actually not the references [16] and [17].

Reply to Question (2): As of now, there are very few reports on the study and practice of collectively recycling PG and RAC into roadbase materials at the same time. For this concern we have added some brief description to the introduction section and rewritten the introduction. The following revisions have been added:

“Phosphogypsum (PG) is mainly the calcium sulfate dihydrate (CaSO4⋅2H2O) formed as a by-product of the production of fertilizer, particularly phosphoric acid [1-3]. According to current statistics, the global stockpile of PG has reached 6 billion tons with an increasing rate of 200 million tons per year [4]. Because of this, large amounts of PG are disposed at the production sites for storage. As of now, the reports are stating that they are recycled and reused in various fields such as chemistry, agriculture, building, etc, but the consumption is still considered very limited [5]. In addition to PG, waste concrete is also one of the most important municipal solid wastes recycled from buildings, roads, and bridges [6-8]. Similarly, its disposal methods are still the landfilling and stockpiling as major, which, as of now, has caused large occupations of land resources and the pollutions of ecological environment [9,10]. Therefore, it is of great importance and urgency to find a new pathway to collectively recycle PG and waste concrete at large scale, for example, in the application of roadbase materials [11-13].

As PG is very susceptible to the moisture and unfriendly to the environment, the application is actually difficult in different areas [14,15]. Generally, the harmless treatment should be considered at first by a series of processes including physical, chemical and calcination methods, and afterwards, some hemihydrate PG (CaSO4⋅0.5H2O) can be obtained [16-18]. Regarding its application in roadbase materials, researchers worldwide have carried out a series of studies. For instance, Mohammad et al. [19] found that partial replacement of Portland cement using PG can still reach an applicable strength requirement in the rigid pavement. Ding et al. [20] concluded that an appropriate amount of PG can be used in the roadbase mix for application, but excessive PG will cause significant damages, especially moisture-induced damage, to the base. Zhang et al. [21] found that adding an appropriate amount of PG (6%) can effectively improve the performance of lime-fly ash-crushed stone roadbase materials, which proved it is feasible to use PG instead of lix-fly-ash to stabilize the gravel and lime in the gravel mixture. The mentioned studies indicated that it has huge potentials to apply PG in the roadbase materials.

As for the research on RCA in roadbase materials [22,23], using RCA as inorganic mixture is an effective way to recycle waste concrete resources at present. Kox et al[24] applied RCA to concrete pavement, and the results showed that the replacement rate of coarse aggregate of 40% would not have adverse effects. It has little effect on concrete aggregate durability and freeze-thaw resistance. Poon et al. [25] used recycled concrete aggregate and crushed clay brick as the base mixture, and 100% recycled aggregate significantly increased the optimal water content and maximum dry density of the mixture. However, by adding crushed clay bricks instead of recycled aggregate, the index of the mixture can be reduced, and the immersed (California Bearing Ratio) CBR value of the mixture is greater than 30%, which meets the requirements of the specification. Under sandy soil, the columns of different lengths are constructed using the compacted RCA structure. Soil pH values at different depths were measured in the experiment. The results show that the leachate of basic RCA can be fully buffered after carbonization. It has been proved that alkaline RCA leachate will not cause corrosion to metal-clad steel culverts in underground soil [26]. However, there are few reports on the high-performance utilization of PG and RCA in road base materials at the same time, and further in-depth research is needed.

To better recycle PG and RCA into the roadbase materials, this study aims to provide a new pathway to realize their application in high performance. The designed process includes the pretreatment of PG by washing and calcination to prepare calcinated PG (CPG) and the use of sodium metasilicate nonahydrate (SMN) to enhance road base materials containing CPG and RCA. Through mix design and the test characterizations such as compressive strength, wet-dry cycle, freeze-thaw cycle and scanning electron microscope, the effects of PG treatment mode, SMN dosage, wet-dry cycle and freeze-thaw cycle on the performance of roadbase materials will be studied, and the traffic bearing capacity and microstructure characteristics of roadbase materials will also be analyzed.”

Line 85: “The research process of this study is shown in Fig. 1.”  This is not the place to locate this figure. It should be in section 2.

Response: Thanks for your comment. As suggested, the “Fig. 1” is now changed to the section 2. Please check it in the revised manuscript.

Line 99: “which were composed of natural aggregates and adhered mortar” were not there any concrete particles?   if that the case and there were just natural aggregates and mortar it is not regular RCA.

Response: Thanks for your comment. As introduced, the RCA used in this study is obtained from the mechanical processing of waste concrete, which, we think, can be simply understood as the concrete particles. We checked lots of literature or reports, for example, saying “Recycled concrete aggregates contain not only the original aggregates, but also hydrated cement paste.”, “recycled aggregates are usually classified as Recycled Concrete Aggregate (RCA) when it is composed mostly of cement-based fragments and natural rocks,”, etc. In addition, we will improve this if this is not appropriate here.

Table 3: Bulk density instead of stack density.  Please, which standards were used for Apparent density porosity. Why there is no values for NFA porosity and water absorption?

Response: Thanks for your comments.

Reply to Question (1): As this study was based on the practice background in China, we referred to the China standard “Standard for Quality Inspection Methods of Sand and Stone for Ordinary Concrete” (JGJ 52-2006) for the mentioned parameters. The following revision has been done:

“According to JGJ 52-2006 [31], the test results of technical indicators related to coarse and fine aggregates are shown in Table 3.”

JGJ 52-2006: Quality Inspection Methods of Sand and Stone for Ordinary Concrete

Reply to Question (2): For your second concern, it needs to indicate that the current study used larger than 4.75mm-RCA particles to totally replace coarse natural aggregates and the natural river sand less than 4.75mm as the NFA. As the water absorption rate and void ratio of RCA are greater than that of natural aggregate, it has a great influence on the optimal water content and maximum dry density of the base mixture. Therefore, additional water needs be added to achieve the optimal water content of the base mixture. NFA does not adsorb additional water and has little effect on the optimal water content of the base material. This is why we didn’t provide the mentioned values.

Line 121: “(by weight of CPG)” if the substitution is based on the weight and the density differs, the volume varies.  Have you considered this for justifying the results?

Response: Thanks for your comment.

Reply to question: Yes. We consider the density changes to analyze the results obtained. As the density of CPG is a little bit less than that of the used natural aggregates, leading to the slightly expanded volume in mixes, the mechanical properties of the mixes, especially compressive strength, will be decreased correspondingly. For this, we consider using the designed method to solve this issue for improving the properties of roadbase materials containing CPG and RCA.

Table 5: The aggregates (RCA: NFA) are replaced by CPG when the particle size distribution is completely different, how much this affects to the properties studied?   The relation RCA: NFA=3:2 is in weight or in volume?  Why did you consider that ratio?

Response: Thanks for your comments.

Reply to Question (1): Good question. The aggregates replaced by CPG will definitely cause the change in particle size distribution and negatively affect the properties studied. But we considered using an additive to enhance the engineering properties of designed roadbase materials containing CPG. For the results obtained, we found that the engineering properties of roadbase materials containing CPG can be reached meeting the technical standard in different highway grades.

Reply to Question (2): The relation RCA: NFA=3:2 is in weight.

Reply to Question (3): According to JTGT F20-2015, the composition of cement stabilized roadbase materials is 5% cement and 95% aggregates, where the coarse aggregate accounts for approximately 60% by weight of total aggregates for a typical aggregate gradation recommended in China. Therefore, our study aimed to replace all the coarse natural aggregates by coarse RCA particles. This is why the RCA content is 57% (95%×60%=57%).

JTGT F20-2015: Technical Guigelines for Construction of Highway RoadBases

For clear understanding, the Table 5 is now updated to:

Table 5. Mix design of roadbase materials containing CPG/RCA.

Item

Cement(%)

CPG(%)

Aggregates(%)

OMC(%)

MDD(kg/m3)

RCA

NFA

0CPG-57RCA

5

0

57

38

6.73

2.272

10CPG-51RCA

5

10

51

34

5.97

2.059

20CPG-45RCA

5

20

45

30

6.17

2.044

30CPG-39RCA

5

30

39

26

6.65

1.999

40CPG-33RCA

5

40

33

22

7.50

1.931

50CPG-27RCA

5

50

27

18

8.44

1.869

Line 164: “standard sample of 28d” was it dried or at the conditions of curing?

Response: Thanks for your comment. The following revision has been done:

“According to JTG E51-2009, six samples were prepared in each group, and after curing for 28d, the cured samples were individually put into the -18°C cryogenic box for the freezing test.”

In section 3 Results and discussion there is no single reference when it is stated, in the guide for authors in Materials journal, “Discussion: Authors should discuss the results and how they can be interpreted in perspective of previous studies and of the working hypotheses.”  No adding any reference in this section diminishes the quality of the paper abruptly.

Response: Thanks for the comments. The following statements have been added:

“On the other hand, the addition of SMN provides better water stability for the mixture, which is similar to the findings of Shen et al. [32].”

  1. Shen W.G.; Zhou M.K. Investigation on the application of steel slag–fly ash–phosphogypsum solidified material as road base material. J Hazard Mater. 2009, 164(1), 99-104.

“The curing effect of SMN can effectively reduce the generation of cracks, enhance the bonding force, and improve the frost resistance, which is consistent with the results of Wang et al. [33]. These results indicated that an appropriate amount of SMN can significantly improve the frost resistance of 20CPG-45RCA, making it superior to the frost resistance of RCA roabase material. The main reason is that the hydration reaction of SMN promotes the formation of a large amount of cementitious material that plays a role in stabilizing CPG and RCA, further reducing the number of internal voids for lowered water erosion damage. The results are in compliance with the findings of Ruan at al. [34] combined bauxite tailings and sodium silicate to fully solidify the magnesium phosphate cement, which produces zeolite-like hydration product sodium silicate aluminate hydrate, made the structure more compact and effectively reducing cracking.”

  1. Wang, Z.H.; Bai, E. Microwave heating efficiency and frost resistance of concrete modified with powder absorbing materials. Constr Build Mater. 2023, 379,131145.
  2. Ruan, W.; Liao, J. Effects of bauxite tailings and sodium silicate on mechanical properties and hydration mechanism of magnesium phosphate cement. Constr Build Mater. 2023, 366: 130055. “This indicated that the addition of SMN remains the unconfined compressive strength at higher level to samples after freeze-thaw cycles. The above results were similar to that provided by Saberian et al. [36]. To summarize, an appropriate amount of SMN can contribute to significantly improving the engineering properties of roadbase materials while the RCA and CPG were overall considered.”
  3. Saberian, M.; Li, J. Effect of freeze–thaw cycles on the resilient moduli and unconfined compressive strength of rubberized recycled concrete aggregate as pavement base/subbase. Transp Geotech. 2021, 27,100477.

“The results are in compliance with the findings of Nasir of et al. [37] incorporating industrial waste-based pastes wherein adequate curing enabled dissolution of precursor materials which consequently led to formation of the dense skeletal matrix thereby yielding enhancement in overall engineering.”

  1. Nasir, M.; Johari, M.A.M. Influence of heat curing period and temperature on the strength of silico-manganese fume-blast furnace slag-based alkali-activated mortar. Constr Build Mater. 2020, 251,118961.

The subsection 3.4 Microstructure analysis could be located before with the purpose to justify the test results values.

Response: Thanks for your comment. For your suggestion it is actually a good way to document the paper from microstructure analysis to macroscopic properties evaluations. But generally, we will first introduce the test results for properties and then give the corresponding analyses from microstructures in order to understand their changing correlation between properties and structure. Therefore, we suggest remaining unchanged to this section. If it is not acceptable, we will revise it anyway after your further review.

The format of the reference list does not match with Materials requirement.

Response: Thanks for your comment. The format of the reference list has been modified to:

“References

  1. Meskini,; Mechnou, I. Environmental investigation on the use of a phosphogypsum-based road material: Radiological and leaching assessment. J Environ Manage. 2023, 345,118597.
  2. Min,; Jueshi, Q. Activation of fly ash–lime systems using calcined phosphogypsum. Constr Build Mater. 2008, 22(5),1004-1008.
  3. Sun,; Li, W. A new eco-friendly concrete made of high content phosphogypsum based aggregates and binder: Mechanical properties and environmental benefits. J Clean Prod. 2023, 400,136555.
  4. Murali,; Azab, M. Recent research in utilization of phosphogypsum as building materials: Review. J Mater Res Technol. 2023, 25,960-987.
  5. Wu,H.; Ren, Y.C. Utilization path of bulk industrial solid waste: A review on the multi-directional resource utilization path of phosphogypsum. J Environ Manage. 2022, 313,114957.
  6. Tejas,; Pasla, D. Assessment of mechanical and durability properties of composite cement-based recycled aggregate concrete. Constr Build Mater. 2023, 387,131620.
  7. Sua-iam,; Makul, N. Self-compacting concrete produced with recycled concrete aggregate coated by a polymer-based agent: A case study. Case Stud Constr Mat. 2023, 19, e02351.
  8. Xu, X.; Luo, Y.; Anand, S. Potential use of recycled concrete aggregate (RCA) for sustainable asphalt pavements of the future: A state-of-the-art review. J Clean Prod. 344 2022,
  9. Hu,; Yang, C. Study on the impermeability of recycled aggregate thermal insulation concrete. J Build Eng. 2023, 77,107400.
  10. Xie,Z.; Zhang, R. The influence of environmental factors and precipitation precursors on enzyme-induced carbonate precipitation (EICP) process and its application on modification of recycled concrete aggregates. J Clean Prod. 2023, 395,136444.
  11. Lal, C.; Jail, S.G. Sustainable development of recycled concrete aggregate through optimized acid-mechanical treatment: A simplified approach. Constr Build Mater. 2023, 399,132559.
  12. Xu, X.; Leng, Sustainable practice in pavement engineering through value-added collective recycling of waste plastic and waste tyre rubber. Engineering-prc. 2021, 7(6), 857-867.
  13. Engelsen,J.; van der Sloot. H.A. Long-term leaching from recycled concrete aggregates applied as sub-base material in road construction. Sci Tot Environ. 2017, 587-588, 94-101.
  14. Gracioli,; Angulski da Luz C. Influence of the calcination temperature of phosphogypsum on the performance of supersulfated cements. Constr Build Mater. 2020, 262,119961.
  15. Cao,X.; Yi, W. Recycling of phosphogypsum to prepare gypsum plaster: Effect of calcination temperature. J Build Eng. 2022, 45,103511.
  16. Geraldo,H.; Costa, A.R.D. Calcination parameters on phosphogypsum waste recycling. Constr Build Mater. 2020, 256, 119406.
  17. Bumanis,; Zorica, J. Technological properties of phosphogypsum binder obtained from fertilizer production waste. Energy Procedia. 2018, 147,301-308.
  18. DeÄŸirmenci, Utilization of phosphogypsum as raw and calcined material in manufacturing of building products. Constr Build Mater. 2008, 22(8),1857-1862.
  19. Mohammad,; Rana, A.S. Incorporation of phosphogypsum with cement in rigid Pavement: An approach towards sustainable development. Materials Today: Proceedings. 2023.
  20. Ding,; Shi, M. Failure of Roadway Subbase Induced by Overuse of Phosphogypsum. Journal of Performance of Constructed Facilities. 2019, 33(2).
  21. Zhang, H.; Cheng, Y.; Yang, Modification of lime-fly ash-crushed stone with phosphogypsum for road base. Adv Civ Eng. 2020, 2020: 1-7.
  22. Engelsen,J.; Wibetoe, G. Field site leaching from recycled concrete aggregates applied as sub-base material in road construction. Sci Tot Environ. 2012, 427-428,86-97.
  23. Rahman,A.; Imteaz, M. Suitability of recycled construction and demolition aggregates as alternative pipe backfilling materials. J Clean Prod. 2014, 66,75-84.
  24. Kox,; Vanroelen, G.l. Experimental evaluation of the high-grade properties of recycled concrete aggregates and their application in concrete road pavement construction. Case Stud Constr Mat. 2019, 11,e00282.
  25. Poon,S.; Chan, D. Feasible use of recycled concrete aggregates and crushed clay brick as unbound road sub-base. Constr Build Mater. 2006, 20(8),578-585.
  26. Oliveira,D.; Gupta, N. pH attenuation by soils underlying recycled concrete aggregate road base. Resour Conserv Recy. 2020, 161,104987.
  27. JGJ 52-2006, Standard for technical requirements and test method of sand and crushed stone (or gravel) for ordinary concrete, China Architecture & Building Press, 2006.
  28. GB 175-1999, Portland Cement and Ordinary Portland Cement, Standards Press of China, 1999.
  29. JTG E51-2009, Test Methods Of Materials Stabilized with Inorganic Binders for Highway Engineering, China Communications Press, 2009.
  30. JTGT F20-2015, Technical Guigelines for Construction of Highway RoadBases, China Communications Press, 2015.
  31. Nasir,; Islam, A.B.M.S. Evolution of Arabic Gum-based green mortar towards enhancing the engineering properties – Fresh, mechanical, and microstructural investigation. Constr Build Mater. 2023, 365,130025.
  32. Shen W.; Zhou M.K. Investigation on the application of steel slag–fly ash–phosphogypsum solidified material as road base material. J Hazard Mater. 2009, 164(1), 99-104.
  33. Wang,H.; Bai, E. Microwave heating efficiency and frost resistance of concrete modified with powder absorbing materials. Constr Build Mater. 2023, 379,131145.
  34. Ruan,; Liao, J. Effects of bauxite tailings and sodium silicate on mechanical properties and hydration mechanism of magnesium phosphate cement. Constr Build Mater. 2023, 366: 130055.
  35. JTG D50-2017, Specifications for Design of Highway Asphalt Pavement, China Communications Press, 2017.
  36. Saberian,; Li, J. Effect of freeze–thaw cycles on the resilient moduli and unconfined compressive strength of rubberized recycled concrete aggregate as pavement base/subbase. Transp Geotech. 2021, 27,100477.
  37. Nasir,; Johari, M.A.M. Influence of heat curing period and temperature on the strength of silico-manganese fume-blast furnace slag-based alkali-activated mortar. Constr Build Mater. 2020, 251,118961.”

Reviewer 2 Report

The introduction needs a little bit of rewriting to emphasize more in the use of PG and RCA as road-base materials. More than half of the citations in the introduction section are related to the environmental impact of PG and construction demolition wastes (concrete) which is only partially related to the topic of the manuscript.

Line 74 - what is the meaning of CBR?

Line 78 - is sodium silicate the same with sodium metasilicate nonahydrate?

Table 2 - is the particle size distribution of CPG similar to PG?

Figure 1 - SEM investigations appears 2 times (please remove one block of the diagram)

Line 116 - CPG replaces NFA? If so, it should be clearly specified in the manuscript.

Line 124 and Table 5 - the naming of the mixes should be clearly explained. How did the authors decided on 57RCS (line 124)? Please include RCA and NFA separately in Table 5 for a more clear picture of the mix.

Section 2.3 - please specify how many samples were considered for each mix in Table 5 and how many were used for each test presented in section 2.3.

Lines 268-270 - please cite the appropriate scientific literature to support this statement or provide experimental data / measurement to support it.

Figure 11 - if possible, please highlight the contours of the aggregates. It is very difficult to discern from the provided photos.

Line 316 - "heavy and extremely heavy traffic"

Please include all the cited sources in the reference list, standards and codes included.

Lines 47-48 - please rephrase. The reader understands NCA are obtained after crushing and screening and not RCA

Line 53 - please replace "disease" by "defects" or an equivalent term

Line 54 - "extensive" instead of "comprehensive"

Line 58 - replace "reflected" by "concluded"

Line 60 - "fly-ash" instead of "flying ash"

Line 61 - what do you understand by "litter"?

Line 63 - "fly ash and PG as partial replacement of cement..."

Lines 81/170 - what do you understand by "lateral compressive strength"?

Table 3 - replace "stack" by "bulk"

Line 116 - "CPG replaced ...." Please specify what type of aggregate was replaced by CPG.

Line 133 - replace "treatment" by "removal"

Lines 147-149 - the paragraph is not clear. Please rephrase.

Line 160 - "unlimited"? I think it is "unconfined"

Lines 165-166 - it is not clear what the 20 mm gap has to do with the freezing time of 16 hours.

Line 176 - the scanning speed is indicated by the wrong unit of measure

Line 284 - I think adding SMN presents an advantage not "disadvantage".

Author Response

Reviewer #2:

The introduction needs a little bit of rewriting to emphasize more in the use of PG and RCA as road-base materials. More than half of the citations in the introduction section are related to the environmental impact of PG and construction demolition wastes (concrete) which is only partially related to the topic of the manuscript.

Response: Thanks for your helpful comment. As suggested, the introduction has been rewritten to:

“Phosphogypsum (PG) is mainly the calcium sulfate dihydrate (CaSO4⋅2H2O) formed as a by-product of the production of fertilizer, particularly phosphoric acid [1-3]. According to current statistics, the global stockpile of PG has reached 6 billion tons with an increasing rate of 200 million tons per year [4]. Because of this, large amounts of PG are disposed at the production sites for storage. As of now, the reports are stating that they are recycled and reused in various fields such as chemistry, agriculture, building, etc, but the consumption is still considered very limited [5]. In addition to PG, waste concrete is also one of the most important municipal solid wastes recycled from buildings, roads, and bridges [6-8]. Similarly, its disposal methods are still the landfilling and stockpiling as major, which, as of now, has caused large occupations of land resources and the pollutions of ecological environment [9,10]. Therefore, it is of great importance and urgency to find a new pathway to collectively recycle PG and waste concrete at large scale, for example, in the application of roadbase materials [11-13].

As PG is very susceptible to the moisture and unfriendly to the environment, the application is actually difficult in different areas [14,15]. Generally, the harmless treatment should be considered at first by a series of processes including physical, chemical and calcination methods, and afterwards, some hemihydrate PG (CaSO4⋅0.5H2O) can be obtained [16-18]. Regarding its application in roadbase materials, researchers worldwide have carried out a series of studies. For instance, Mohammad et al. [19] found that partial replacement of Portland cement using PG can still reach an applicable strength requirement in the rigid pavement. Ding et al. [20] concluded that an appropriate amount of PG can be used in the roadbase mix for application, but excessive PG will cause significant damages, especially moisture-induced damage, to the base. Zhang et al. [21] found that adding an appropriate amount of PG (6%) can effectively improve the performance of lime-fly ash-crushed stone roadbase materials, which proved it is feasible to use PG instead of lix-fly-ash to stabilize the gravel and lime in the gravel mixture. The mentioned studies indicated that it has huge potentials to apply PG in the roadbase materials.

As for the research on RCA in roadbase materials [22,23], using RCA as inorganic mixture is an effective way to recycle waste concrete resources at present. Kox et al[24] applied RCA to concrete pavement, and the results showed that the replacement rate of coarse aggregate of 40% would not have adverse effects. It has little effect on concrete aggregate durability and freeze-thaw resistance. Poon et al. [25] used recycled concrete aggregate and crushed clay brick as the base mixture, and 100% recycled aggregate significantly increased the optimal water content and maximum dry density of the mixture. However, by adding crushed clay bricks instead of recycled aggregate, the index of the mixture can be reduced, and the immersed (California Bearing Ratio) CBR value of the mixture is greater than 30%, which meets the requirements of the specification. Under sandy soil, the columns of different lengths are constructed using the compacted RCA structure. Soil pH values at different depths were measured in the experiment. The results show that the leachate of basic RCA can be fully buffered after carbonization. It has been proved that alkaline RCA leachate will not cause corrosion to metal-clad steel culverts in underground soil [26]. However, there are few reports on the high-performance utilization of PG and RCA in road base materials at the same time, and further in-depth research is needed.

To better recycle PG and RCA into the roadbase materials, this study aims to provide a new pathway to realize their application in high performance. The designed process includes the pretreatment of PG by washing and calcination to prepare calcinated PG (CPG) and the use of sodium metasilicate nonahydrate (SMN) to enhance road base materials containing CPG and RCA. Through mix design and the test characterizations such as compressive strength, wet-dry cycle, freeze-thaw cycle and scanning electron microscope, the effects of PG treatment mode, SMN dosage, wet-dry cycle and freeze-thaw cycle on the performance of roadbase materials will be studied, and the traffic bearing capacity and microstructure characteristics of roadbase materials will also be analyzed.”

Line 74 - what is the meaning of CBR?

Response: Thanks for your comment. The full name of CBR has been added at its first appearance in the rewritten introduction. The following sentence can be checked:

“…Adding PG to natural soil can increase unconfined compressive strength (UCS) and California Bearing Ratio (CBR). ...”

Line 78 - is sodium silicate the same with sodium metasilicate nonahydrate?

Response: Thanks for your careful review. Actually, the accurate name is the latter one. We have revised the name in the following sentence:

“In this study, it is proposed to pretreat PG by washing and calcination, and use sodium metasilicate nonahydrate (SMN) to enhance the treatment of cement stable calcination phosphorus gypsum (CPG) and RCA road base materials.”

Table 2 - is the particle size distribution of CPG similar to PG?

Response: Thanks for your comment. Actually, Table 2 gives the particle size distribution of CPG. As the collected PG has high absorption, it is not easily to go through the sieving process for accurate results of particle size distribution. Sorry for our carelessness. We have revised the table to:

Table 2. Particle size distribution of CPG

Particle size/mm

2.36

1.18

0.6

0.3

0.15

0.075

Passing percentage/%

100

96.87

93.26

90.89

78.36

9.32

Figure 1 - SEM investigations appears 2 times (please remove one block of the diagram)

Response: Thanks for your careful review. According to the suggestion, the flowchart figure has been modified to:

Figure 4. The research flow chart of this study.

Line 116 - CPG replaces NFA?  If so, it should be clearly specified in the manuscript.

Response: Thanks for your comment. All graded aggregates, including RCA and NFA, are replaced by CPG in weight with proportions. This design change will definitely affect the original aggregate gradation. In spite of this, we consider adopting a new technical method to enhance the engineering properties of these gradation-changed roadbase materials containing CPG. Satisfactorily, from the 7d compressive strength results, the target roadbase materials can meet different application requirements of highway grades, even reaching an extremely traffic grade.

The following changes for clear understanding:

Table 5. Mix design of roadbase materials containing CPG/RCA.

Item

Cement(%)

CPG(%)

Aggregates(%)

OMC(%)

MDD(kg/m3)

RCA

NFA

0CPG-57RCA

5

0

57

38

6.73

2.272

10CPG-51RCA

5

10

51

34

5.97

2.059

20CPG-45RCA

5

20

45

30

6.17

2.044

30CPG-39RCA

5

30

39

26

6.65

1.999

40CPG-33RCA

5

40

33

22

7.50

1.931

50CPG-27RCA

5

50

27

18

8.44

1.869

Line 124 and Table 5 - the naming of the mixes should be clearly explained.  How did the authors decided on 57RCS (line 124)?  Please include RCA and NFA separately in Table 5 for a more clear picture of the mix.

Response: Thanks for your comment. As suggested, for the clear understanding of the mixes, the Table 5 is now changed to:

Table 5. Mix design of roadbase materials containing CPG/RCA.

Item

Cement(%)

CPG(%)

Aggregates(%)

OMC(%)

MDD(kg/m3)

RCA

NFA

0CPG-57RCA

5

0

57

38

6.73

2.272

10CPG-51RCA

5

10

51

34

5.97

2.059

20CPG-45RCA

5

20

45

30

6.17

2.044

30CPG-39RCA

5

30

39

26

6.65

1.999

40CPG-33RCA

5

40

33

22

7.50

1.931

50CPG-27RCA

5

50

27

18

8.44

1.869

Section 2.3 - please specify how many samples were considered for each mix in Table 5 and how many were used for each test presented in section 2.3.

Response: Thanks for your comments. The following descriptions or changes have been added:

Section 2.2.1: “At the same time, the maximum dry density (MDD) and optimal moisture content (OMC) were calculated by the test results of six samples for each group, according to JTG E51-2009[29], as shown in Table 5.”

Section 2.3.1: “According to JTG E51-2009 , the Φ100mm×100mm cylindrical samples were molded by static pressure and the cured for 7d, and six samples in each group were prepared for the test.”

Section 2.3.2: “The whole process is regarded as a complete wetting and drying cycle. Six samples were prepared for test in each group.”

Section 2.3.3: “According to JTG E51-2009, six samples were prepared in each group, and after curing for 28d, the cured samples were individually put into the -18°C cryogenic box for a 16h-freezing test.”

Lines 268-270 - please cite the appropriate scientific literature to support this statement or provide experimental data / measurement to support it.

Response: Thanks for the helpful suggestion. Appropriate scientific literature has been cited in this section, Please check below:

 “The results are in compliance with the findings of Ruan at al. [34] combined bauxite tailings and sodium silicate to fully solidify the magnesium phosphate cement, which produces zeolite-like hydration product sodium silicate aluminate hydrate, made the structure more compact and effectively reducing cracking.”

  1. Ruan, W.; Liao, J. Effects of bauxite tailings and sodium silicate on mechanical properties and hydration mechanism of magnesium phosphate cement. Constr Build Mater. 2023, 366: 130055.

Figure 11 - if possible, please highlight the contours of the aggregates.  It is very difficult to discern from the provided photos.

Response: Thanks for your comment. As these microstructural images were captured at 5kx, the coarse RCA aggregates were not easily presented at located regions, and fine aggregates were mixed into the cement paste. Therefore, it is impossible to highlight the contours of the aggregates.

Actually, these SEM images are used to analyze if the proposed technical method can repair the cracks caused by the incorporation of CPG particles.

Line 316 - "heavy and extremely heavy traffic".

Response: Thanks for your comment. After checking the results, the following revisions have been done:

“…, which meets the requirement of extremely and very heavy traffic grade.”

Please include all the cited sources in the reference list, standards and codes included.

Response: Thanks for your helpful suggestion. The following changes have been made:

“References

  1. Meskini,; Mechnou, I. Environmental investigation on the use of a phosphogypsum-based road material: Radiological and leaching assessment. J Environ Manage. 2023, 345,118597.
  2. Min,; Jueshi, Q. Activation of fly ash–lime systems using calcined phosphogypsum. Constr Build Mater. 2008, 22(5),1004-1008.
  3. Sun,; Li, W. A new eco-friendly concrete made of high content phosphogypsum based aggregates and binder: Mechanical properties and environmental benefits. J Clean Prod. 2023, 400,136555.
  4. Murali,; Azab, M. Recent research in utilization of phosphogypsum as building materials: Review. J Mater Res Technol. 2023, 25,960-987.
  5. Wu,H.; Ren, Y.C. Utilization path of bulk industrial solid waste: A review on the multi-directional resource utilization path of phosphogypsum. J Environ Manage. 2022, 313,114957.
  6. Tejas,; Pasla, D. Assessment of mechanical and durability properties of composite cement-based recycled aggregate concrete. Constr Build Mater. 2023, 387,131620.
  7. Sua-iam,; Makul, N. Self-compacting concrete produced with recycled concrete aggregate coated by a polymer-based agent: A case study. Case Stud Constr Mat. 2023, 19, e02351.
  8. Xu, X.; Luo, Y.; Anand, S. Potential use of recycled concrete aggregate (RCA) for sustainable asphalt pavements of the future: A state-of-the-art review. J Clean Prod. 344 2022,
  9. Hu,; Yang, C. Study on the impermeability of recycled aggregate thermal insulation concrete. J Build Eng. 2023, 77,107400.
  10. Xie,Z.; Zhang, R. The influence of environmental factors and precipitation precursors on enzyme-induced carbonate precipitation (EICP) process and its application on modification of recycled concrete aggregates. J Clean Prod. 2023, 395,136444.
  11. Lal, C.; Jail, S.G. Sustainable development of recycled concrete aggregate through optimized acid-mechanical treatment: A simplified approach. Constr Build Mater. 2023, 399,132559.
  12. Xu, X.; Leng, Sustainable practice in pavement engineering through value-added collective recycling of waste plastic and waste tyre rubber. Engineering-prc. 2021, 7(6), 857-867.
  13. Engelsen,J.; van der Sloot. H.A. Long-term leaching from recycled concrete aggregates applied as sub-base material in road construction. Sci Tot Environ. 2017, 587-588, 94-101.
  14. Gracioli,; Angulski da Luz C. Influence of the calcination temperature of phosphogypsum on the performance of supersulfated cements. Constr Build Mater. 2020, 262,119961.
  15. Cao,X.; Yi, W. Recycling of phosphogypsum to prepare gypsum plaster: Effect of calcination temperature. J Build Eng. 2022, 45,103511.
  16. Geraldo,H.; Costa, A.R.D. Calcination parameters on phosphogypsum waste recycling. Constr Build Mater. 2020, 256, 119406.
  17. Bumanis,; Zorica, J. Technological properties of phosphogypsum binder obtained from fertilizer production waste. Energy Procedia. 2018, 147,301-308.
  18. DeÄŸirmenci, Utilization of phosphogypsum as raw and calcined material in manufacturing of building products. Constr Build Mater. 2008, 22(8),1857-1862.
  19. Mohammad,; Rana, A.S. Incorporation of phosphogypsum with cement in rigid Pavement: An approach towards sustainable development. Materials Today: Proceedings. 2023.
  20. Ding,; Shi, M. Failure of Roadway Subbase Induced by Overuse of Phosphogypsum. Journal of Performance of Constructed Facilities. 2019, 33(2).
  21. Zhang, H.; Cheng, Y.; Yang, Modification of lime-fly ash-crushed stone with phosphogypsum for road base. Adv Civ Eng. 2020, 2020: 1-7.
  22. Engelsen,J.; Wibetoe, G. Field site leaching from recycled concrete aggregates applied as sub-base material in road construction. Sci Tot Environ. 2012, 427-428,86-97.
  23. Rahman,A.; Imteaz, M. Suitability of recycled construction and demolition aggregates as alternative pipe backfilling materials. J Clean Prod. 2014, 66,75-84.
  24. Kox,; Vanroelen, G.l. Experimental evaluation of the high-grade properties of recycled concrete aggregates and their application in concrete road pavement construction. Case Stud Constr Mat. 2019, 11,e00282.
  25. Poon,S.; Chan, D. Feasible use of recycled concrete aggregates and crushed clay brick as unbound road sub-base. Constr Build Mater. 2006, 20(8),578-585.
  26. Oliveira,D.; Gupta, N. pH attenuation by soils underlying recycled concrete aggregate road base. Resour Conserv Recy. 2020, 161,104987.
  27. JGJ 52-2006, Standard for technical requirements and test method of sand and crushed stone (or gravel) for ordinary concrete, China Architecture & Building Press, 2006.
  28. GB 175-1999, Portland Cement and Ordinary Portland Cement, Standards Press of China, 1999.
  29. JTG E51-2009, Test Methods Of Materials Stabilized with Inorganic Binders for Highway Engineering, China Communications Press, 2009.
  30. JTGT F20-2015, Technical Guigelines for Construction of Highway RoadBases, China Communications Press, 2015.
  31. Nasir,; Islam, A.B.M.S. Evolution of Arabic Gum-based green mortar towards enhancing the engineering properties – Fresh, mechanical, and microstructural investigation. Constr Build Mater. 2023, 365,130025.
  32. Shen W.; Zhou M.K. Investigation on the application of steel slag–fly ash–phosphogypsum solidified material as road base material. J Hazard Mater. 2009, 164(1), 99-104.
  33. Wang,H.; Bai, E. Microwave heating efficiency and frost resistance of concrete modified with powder absorbing materials. Constr Build Mater. 2023, 379,131145.
  34. Ruan,; Liao, J. Effects of bauxite tailings and sodium silicate on mechanical properties and hydration mechanism of magnesium phosphate cement. Constr Build Mater. 2023, 366: 130055.
  35. JTG D50-2017, Specifications for Design of Highway Asphalt Pavement, China Communications Press, 2017.
  36. Saberian,; Li, J. Effect of freeze–thaw cycles on the resilient moduli and unconfined compressive strength of rubberized recycled concrete aggregate as pavement base/subbase. Transp Geotech. 2021, 27,100477.
  37. Nasir,; Johari, M.A.M. Influence of heat curing period and temperature on the strength of silico-manganese fume-blast furnace slag-based alkali-activated mortar. Constr Build Mater. 2020, 251,118961.”

Comments on the Quality of English Language.

Lines 47-48 - please rephrase. The reader understands NCA are obtained after crushing and screening and not RCA

Line 53 - please replace "disease" by "defects" or an equivalent term

Line 54 - "extensive" instead of "comprehensive"

Line 58 - replace "reflected" by "concluded"

Line 60 - "fly-ash" instead of "flying ash"

Line 61 - what do you understand by "litter"?

Line 63 - "fly ash and PG as partial replacement of cement..."

Lines 81/170 - what do you understand by "lateral compressive strength"?

Table 3 - replace "stack" by "bulk"

Line 116 - "CPG replaced ...." Please specify what type of aggregate was replaced by CPG.

Line 133 - replace "treatment" by "removal"

Lines 147-149 - the paragraph is not clear. Please rephrase.

Line 160 - "unlimited"? I think it is "unconfined"

Lines 165-166 - it is not clear what the 20 mm gap has to do with the freezing time of 16 hours.

Line 176 - the scanning speed is indicated by the wrong unit of measure

Line 284 - I think adding SMN presents an advantage not "disadvantage"

Response: Thanks for your careful reading. Except for these language revisions, all others we found have been revised as well for the language improvement. Relevant changes please find in the revised manuscript.

Line 116 - "CPG replaced ...." Please specify what type of aggregate was replaced by CPG.

Response: Thanks for your suggestion. As mentioned to previous comment, all graded aggregates, including RCA and NFA, are replaced by CPG in weight with proportions. Please check the changes in Table 5.

Table 5. Mix design of roadbase materials containing CPG/RCA.

Item

Cement(%)

CPG(%)

Aggregates(%)

OMC(%)

MDD(kg/m3)

RCA

NFA

0CPG-57RCA

5

0

57

38

6.73

2.272

10CPG-51RCA

5

10

51

34

5.97

2.059

20CPG-45RCA

5

20

45

30

6.17

2.044

30CPG-39RCA

5

30

39

26

6.65

1.999

40CPG-33RCA

5

40

33

22

7.50

1.931

50CPG-27RCA

5

50

27

18

8.44

1.869

The following language revisions were done:

“The graded aggregates, including RCA and NFA, were replaced by CPG at 10%, 20%, 30%, 40%, and 50% by weight, respectively.”

Lines 147-149 - the paragraph is not clear. Please rephrase.

Response: Thanks for your comment. The following changes have been made:

“According to JTGT F20-2015[30], the traffic carrying grade of PG/RCA roadbase materials were evaluated and divided based on the 7d unconfined compressive strength result, as shown in Table 7.

  1. JTGT F20-2015, Technical Guigelines for Construction of Highway RoadBases, China Communications Press, 2015.

Lines 165-166 - it is not clear what the 20 mm gap has to do with the freezing time of 16 hours.

Response: Thanks for your comment. The following revision has been done:

“According to JTG E51-2009, six samples were prepared in each group, and after curing for 28d, the cured samples were individually put into the -18°C cryogenic box for a 16h-freezing test.”

  1. JTG E51-2009, Test Methods Of Materials Stabilized with Inorganic Binders for Highway Engineering, China Communica-tions Press, 2009.

Line 284 - I think adding SMN presents an advantage not "disadvantage"

Response: Thanks for your comment. The following changes have been made:

 “This indicated that the addition of SMN remains the unconfined compressive strength at higher level to samples after freeze-thaw cycles. The above results were similar to that provided by Saberian et al. [36]. To summarize, an appropriate amount of SMN can contribute to significantly improving the engineering properties of roadbase materials while the RCA and CPG were overall considered.”

Reviewer 3 Report

This study focuses on finding sustainable ways to reuse phosphogypsum (PG) by-products from phosphoric acid production and recycled concrete aggregate (RCA) from waste concrete in road construction. The approach involves converting PG into calcinated PG (CPG) through washing and calcination and using sodium metasilicate nonahydrate (SMN) to strengthen cement-stabilized CPG and RCA roadbase materials. Various experiments were conducted, including tests for unconfined compressive strength, resistance to environmental cycling, and microstructure analysis. The results indicate that by using 20% CPG and 11% SMN, the roadbase material meets the requirements for heavy traffic loads with a 7-day unconfined compressive strength of 5.34 MPa. Moreover, the addition of 11% SMN significantly improves the material's resistance to moisture and frost, reduces crack width, and enhances microstructure. This research provides valuable insights for developing more durable roadbase materials

The article is well written, and the experiments are well designed and prove the effectiveness of the material for the intended application.

However, before the article is accepted for publication, I suggest some modifications and clarifications.

1) The use of phosphogypsum in construction materials is quite old, including for use in pavements. Therefore, the authors must make it very clear what innovation this article brings to the existing literature. Also, given the number of articles published about this waste in this type of application, I consider it important to focus, in the introduction, on a slightly more comprehensive review in this regard, demonstrating the gaps in knowledge that this article aims to fill.

2) SMN was undoubtedly the enabler of the use of waste. Although the authors mentioned the formation of CSH in greater quantities, it is necessary to expand the possible mechanisms involved, including references and bibliographical basis.

3) The values 57, 51, 45, 39, and 27 in Table 5 need to be explained, as well as the reason for the choice. Is 57 the standard? If yes, I need to clarify.

4) What is the chemical composition of CPG?

Minor comments:

1) There are a large number of abbreviations, so, if possible, include a list of abbreviations to make reading easier

2) Figures 5 to 9, explain whether the points refer to the mean, and number of experiments carried out, and insert a standard deviation

no comments

Author Response

Reviewer #3:

  1. The use of phosphogypsum in construction materials is quite old, including for use in pavements. Therefore, the authors must make it very clear what innovation this article brings to the existing literature.  Also, given the number of articles published about this waste in this type of application, I consider it important to focus, in the introduction, on a slightly more comprehensive review in this regard, demonstrating the gaps in knowledge that this article aims to fill.

Response: Thanks for your comment. The introduction has been rewritten, please check below:

“Phosphogypsum (PG) is mainly the calcium sulfate dihydrate (CaSO4⋅2H2O) formed as a by-product of the production of fertilizer, particularly phosphoric acid [1-3]. According to current statistics, the global stockpile of PG has reached 6 billion tons with an increasing rate of 200 million tons per year [4]. Because of this, large amounts of PG are disposed at the production sites for storage. As of now, the reports are stating that they are recycled and reused in various fields such as chemistry, agriculture, building, etc, but the consumption is still considered very limited [5]. In addition to PG, waste concrete is also one of the most important municipal solid wastes recycled from buildings, roads, and bridges [6-8]. Similarly, its disposal methods are still the landfilling and stockpiling as major, which, as of now, has caused large occupations of land resources and the pollutions of ecological environment [9,10]. Therefore, it is of great importance and urgency to find a new pathway to collectively recycle PG and waste concrete at large scale, for example, in the application of roadbase materials [11-13].

As PG is very susceptible to the moisture and unfriendly to the environment, the application is actually difficult in different areas [14,15]. Generally, the harmless treatment should be considered at first by a series of processes including physical, chemical and calcination methods, and afterwards, some hemihydrate PG (CaSO4⋅0.5H2O) can be obtained [16-18]. Regarding its application in roadbase materials, researchers worldwide have carried out a series of studies. For instance, Mohammad et al. [19] found that partial replacement of Portland cement using PG can still reach an applicable strength requirement in the rigid pavement. Ding et al. [20] concluded that an appropriate amount of PG can be used in the roadbase mix for application, but excessive PG will cause significant damages, especially moisture-induced damage, to the base. Zhang et al. [21] found that adding an appropriate amount of PG (6%) can effectively improve the performance of lime-fly ash-crushed stone roadbase materials, which proved it is feasible to use PG instead of lix-fly-ash to stabilize the gravel and lime in the gravel mixture. The mentioned studies indicated that it has huge potentials to apply PG in the roadbase materials.

As for the research on RCA in roadbase materials [22,23], using RCA as inorganic mixture is an effective way to recycle waste concrete resources at present. Kox et al[24] applied RCA to concrete pavement, and the results showed that the replacement rate of coarse aggregate of 40% would not have adverse effects. It has little effect on concrete aggregate durability and freeze-thaw resistance. Poon et al. [25] used recycled concrete aggregate and crushed clay brick as the base mixture, and 100% recycled aggregate significantly increased the optimal water content and maximum dry density of the mixture. However, by adding crushed clay bricks instead of recycled aggregate, the index of the mixture can be reduced, and the immersed (California Bearing Ratio) CBR value of the mixture is greater than 30%, which meets the requirements of the specification. Under sandy soil, the columns of different lengths are constructed using the compacted RCA structure. Soil pH values at different depths were measured in the experiment. The results show that the leachate of basic RCA can be fully buffered after carbonization. It has been proved that alkaline RCA leachate will not cause corrosion to metal-clad steel culverts in underground soil [26]. However, there are few reports on the high-performance utilization of PG and RCA in road base materials at the same time, and further in-depth research is needed.

To better recycle PG and RCA into the roadbase materials, this study aims to provide a new pathway to realize their application in high performance. The designed process includes the pretreatment of PG by washing and calcination to prepare calcinated PG (CPG) and the use of sodium metasilicate nonahydrate (SMN) to enhance road base materials containing CPG and RCA. Through mix design and the test characterizations such as compressive strength, wet-dry cycle, freeze-thaw cycle and scanning electron microscope, the effects of PG treatment mode, SMN dosage, wet-dry cycle and freeze-thaw cycle on the performance of roadbase materials will be studied, and the traffic bearing capacity and microstructure characteristics of roadbase materials will also be analyzed.”

  1. SMN was undoubtedly the enabler of the use of waste. Although the authors mentioned the formation of CSH in greater quantities, it is necessary to expand the possible mechanisms involved, including references and bibliographical basis.

Response: Thanks for your comment. Based on your suggestions, some possible mechanism analyses with references are added to:

“On the other hand, the addition of SMN provides better water stability for the mixture, which is similar to the findings of Shen et al.[36] and Silva at al. [20].”

  1. Silva M.V.; de Rezende L.R. et al Phosphogypsum, tropical soil and cement mixtures for asphalt pavements under wet and dry environmental conditions. Resources, Conservation and Recycling. 2019, 144,123-136; DOI:10.1016/j.resconrec.2019.01.029.
  2. Shen W.G.; Zhou M.K. et al Investigation on the application of steel slag–fly ash–phosphogypsum solidified material as road base material. Journal of Hazardous Materials. 2009, 164(1),99-104; DOI:10.1016/j.jhazmat.2008.07.125.

“After freeze-thaw cycle, the proportion of micro-cracks and harmful pores in the base material increases. Although the addition of CPG can fill the pores, it will reduce the cohesiveness of the cement paste. The curing effect of SMN can effectively reduce the generation of cracks, enhance the bonding force, and improve the frost resistance, which is consistent with the results of Wang et al. [37].”

  1. Wang Z.H.; Bai E.L et al Microwave heating efficiency and frost resistance of concrete modified with powder absorbing materials. Construction and Building Materials. 2023, 379,131145; DOI:10.1016/j.conbuildmat.2023.131145.

“the addition of SMN consistently provided the highest unconfined compressive strength for samples after freeze-thaw cycles, and these results were similar to those of Saberian, M et al.[39]”

  1. Saberian M.; Li J. Effect of freeze–thaw cycles on the resilient moduli and unconfined compressive strength of rubberized recycled concrete aggregate as pavement base/subbase. Transportation Geotechnics. 2021, 27,100477; DOI:10.1016/j.trgeo.2020.100477.

“The addition of SMN promoted the hydration process to fully cure the PG/RCA base material, greatly improved the bonding between the skeleton and PG, and significantly reduced the development of cracks. These results are consistent with the findings of Nasir et al. [40] that SMN is beneficial to effectively fill the cracks and pores of the base material caused by the incorporation of CPG and improve its density.”

  1. Nasir M.; Johari M.A.M. et al Influence of heat curing period and temperature on the strength of silico-manganese fume-blast furnace slag-based alkali-activated mortar. Construction and Building Materials. 2020, 251,118961; DOI:10.1016/j.conbuildmat.2020.118961.

  1. The values 57, 51, 45, 39, and 27 in Table 5 need to be explained, as well as the reason for the choice. Is 57 the standard?  If yes, I need to clarify.

Response: Thanks for your valuable comments. For clear understanding, the Table 5 has been changed to:

Table 5. Mix design of roadbase materials containing CPG/RCA.

Item

Cement(%)

CPG(%)

Aggregates(%)

OMC(%)

MDD(kg/m3)

RCA

NFA

0CPG-57RCA

5

0

57

38

6.73

2.272

10CPG-51RCA

5

10

51

34

5.97

2.059

20CPG-45RCA

5

20

45

30

6.17

2.044

30CPG-39RCA

5

30

39

26

6.65

1.999

40CPG-33RCA

5

40

33

22

7.50

1.931

50CPG-27RCA

5

50

27

18

8.44

1.869

Reply to question: 57 is not the standard recommendation. According to JTGT F20-2015, the composition of cement stabilized roadbase materials is 5% cement and 95% aggregates, where the coarse aggregate accounts for 60% by weight of total aggregates for a typical aggregate gradation. Therefore, our study aimed to replace all the coarse natural aggregates by coarse RCA particles. This is why the RCA content is 57% (95%×60%=57%).

  1. What is the chemical composition of CPG.

Response: Thanks for your question. We didn’t test the chemical composition of CPG, as we considered whether the engineering properties of roadbase materials containing CPG can meet the technical requirement for practice as our first task. As you mentioned, we will test it in the future for the further continued study. Thanks for your understanding.

  1. There are a large number of abbreviations, so, if possible, include a list of abbreviations to make reading easier.

Response: Thanks for your comments. Please check below for a list of abbreviations.

Table 6. Summary of name abbreviations.

Name

Name abbreviation

Phosphopypsum

PG

Calcination phosphopypsum

CPG

Sodium metasilicate nonahydrate

SMN

Rcycled concrete aggregate

RCA

Natural aggregate

NA

Natural coarse aggregate

NCA

Natural fine aggregate

NFA

  1. Figures 5 to 9, explain whether the points refer to the mean, and number of experiments carried out, and insert a standard deviation.

Response: Thanks for your comment. Figures 5 to 9: the figure points refer to average values, calculated from the test results of six samples. The figures have been modified to:

Figure 5. 7d and 28d unconfined compressive strength of different CPG/RCA road base materials.

Figure 6. Effect of SMN contents on 7d unconfined compressive strength of 20CPG-45RCA roabase material.

Figure 7. Unconfined compressive strength of different PG/RCA roadbase materials.

Figure 8. Effect of wet-dry cycle on the 7d unconfined compressive strength of PG/RCA roadbase material.

Figure 9. The effect of freeze-thaw cycles on the unconfined compressive strength of different PG/RCA roadbase materials cured for 28d.

Reviewer 4 Report

The research article titled "Performance improvement and microstructure characterization 2 of cement stabilized roadbase materials containing phos-3 phogypsum/recycled concrete aggregate" is interesting and well-written. The authors assessed the potency of some waste materials and found beneficial in enhancing the properties of roadbase by conducting series of tests. The article can indeed add value to the literature. However, there are some minor to major shortcomings in the article which need to be addressed prior to acceptance. Please find attached pdf file for my detailed 12 comments.

Author Response

Reviewer #4:

  1. Please provide reference.

Response: The reference has been cited to the corresponding position to support the sentence mentioned.

  1. Please add some literature which explored the impact of washing and calcination to enhance the properties of the materials. For instance, which materials have been used so far, what range of temperature/period and others studied, what was the optimum condition, etc.

Response: We thank you for your careful reading. The introduction has been rewritten with some updated literature. Please check below:

“Phosphogypsum (PG) is mainly the calcium sulfate dihydrate (CaSO4⋅2H2O) formed as a by-product of the production of fertilizer, particularly phosphoric acid [1-3]. According to current statistics, the global stockpile of PG has reached 6 billion tons with an increasing rate of 200 million tons per year [4]. Because of this, large amounts of PG are disposed at the production sites for storage. As of now, the reports are stating that they are recycled and reused in various fields such as chemistry, agriculture, building, etc, but the consumption is still considered very limited [5]. In addition to PG, waste concrete is also one of the most important municipal solid wastes recycled from buildings, roads, and bridges [6-8]. Similarly, its disposal methods are still the landfilling and stockpiling as major, which, as of now, has caused large occupations of land resources and the pollutions of ecological environment [9,10]. Therefore, it is of great importance and urgency to find a new pathway to collectively recycle PG and waste concrete at large scale, for example, in the application of roadbase materials [11-13].

As PG is very susceptible to the moisture and unfriendly to the environment, the application is actually difficult in different areas [14,15]. Generally, the harmless treatment should be considered at first by a series of processes including physical, chemical and calcination methods, and afterwards, some hemihydrate PG (CaSO4⋅0.5H2O) can be obtained [16-18]. Regarding its application in roadbase materials, researchers worldwide have carried out a series of studies. For instance, Mohammad et al. [19] found that partial replacement of Portland cement using PG can still reach an applicable strength requirement in the rigid pavement. Ding et al. [20] concluded that an appropriate amount of PG can be used in the roadbase mix for application, but excessive PG will cause significant damages, especially moisture-induced damage, to the base. Zhang et al. [21] found that adding an appropriate amount of PG (6%) can effectively improve the performance of lime-fly ash-crushed stone roadbase materials, which proved it is feasible to use PG instead of lix-fly-ash to stabilize the gravel and lime in the gravel mixture. The mentioned studies indicated that it has huge potentials to apply PG in the roadbase materials.

As for the research on RCA in roadbase materials [22,23], using RCA as inorganic mixture is an effective way to recycle waste concrete resources at present. Kox et al[24] applied RCA to concrete pavement, and the results showed that the replacement rate of coarse aggregate of 40% would not have adverse effects. It has little effect on concrete aggregate durability and freeze-thaw resistance. Poon et al. [25] used recycled concrete aggregate and crushed clay brick as the base mixture, and 100% recycled aggregate significantly increased the optimal water content and maximum dry density of the mixture. However, by adding crushed clay bricks instead of recycled aggregate, the index of the mixture can be reduced, and the immersed (California Bearing Ratio) CBR value of the mixture is greater than 30%, which meets the requirements of the specification. Under sandy soil, the columns of different lengths are constructed using the compacted RCA structure. Soil pH values at different depths were measured in the experiment. The results show that the leachate of basic RCA can be fully buffered after carbonization. It has been proved that alkaline RCA leachate will not cause corrosion to metal-clad steel culverts in underground soil [26]. However, there are few reports on the high-performance utilization of PG and RCA in road base materials at the same time, and further in-depth research is needed.

To better recycle PG and RCA into the roadbase materials, this study aims to provide a new pathway to realize their application in high performance. The designed process includes the pretreatment of PG by washing and calcination to prepare calcinated PG (CPG) and the use of sodium metasilicate nonahydrate (SMN) to enhance road base materials containing CPG and RCA. Through mix design and the test characterizations such as compressive strength, wet-dry cycle, freeze-thaw cycle and scanning electron microscope, the effects of PG treatment mode, SMN dosage, wet-dry cycle and freeze-thaw cycle on the performance of roadbase materials will be studied, and the traffic bearing capacity and microstructure characteristics of roadbase materials will also be analyzed.”

  1. Could you please report the standards which were used to assess the properties in Tables 3 and 4?.

Response: Thanks for your comment. The following statements have been added:

 “According to JGJ 52-2006 [27], the test results of technical indicators related to coarse and fine aggregates are shown in Table 3.”

  1. JGJ 52-2006, Standard for technical requirements and test method of sand and crushed stone (or gravel) for ordinary concrete, China Architecture & Building Press, 2006.

“Ordinary Portland cement is a locally supplied P.O42.5 grade product, and the performance results are tested and shown in Table 4, according to GB 175-1999 [28].”

  1. GB 175-1999, Portland Cement and Ordinary Portland Cement, Standards Press of China, 1999.

  1. Please explain what was the rational of selection of this temperature and period?

Response: Thanks for your comment. Maybe your question is on why we chose the designed temperature and time to treat PG for preparing CPG. As per lots of current studies and our preliminary TG-DSC tests, we confirm that this disposal condition can remove 1.5H2O from CaSO4·2H2O in PG for realizing the calcination. Here, we add a citation to support this sentence. The following change has been done:

  1. Please report the mix proportion in kg/m3.

Response: Thanks for your comment. As other reviewers’ comments, we have made this table clearer for better understanding. If it is not satisfied, we will revise this further. Thanks for your understanding. Please check the changed table below:

Table 5. Mix design of roadbase materials containing CPG/RCA.

Item

Cement(%)

CPG(%)

Aggregates(%)

OMC(%)

MDD(kg/m3)

RCA

NFA

0CPG-57RCA

5

0

57

38

6.73

2.272

10CPG-51RCA

5

10

51

34

5.97

2.059

20CPG-45RCA

5

20

45

30

6.17

2.044

30CPG-39RCA

5

30

39

26

6.65

1.999

40CPG-33RCA

5

40

33

22

7.50

1.931

50CPG-27RCA

5

50

27

18

8.44

1.869

  1. The methodology of SEM analysis is not accurate and clear! The authors may go through the following recent and relevant paper and may refer to support:

[26]https://doi.org/10.1016/j.conbuildmat.2022.130025.

Response: Thanks for your comment. This paper is well for reference and has been cited. Section 2.3.4 has been rewritten to:

“The PG/RCA base material samples cured for 28d were cut, and rectangular samples no larger than 5mm were taken, and the microstructure of the cut samples was analyzed by SEM machine. First, the sample was gold-plated to study the microstructure changes, and then the following conditions were set: high vacuummode, physical working distance of ~10 mm, an accelerated volage of 20.00 kV[35]. The final image is enlarged between 100 and 10,000 times, with a scale of 1mm to 1µm.”

  1. There is no citation in the results section! Further, more insights are required in this section. The authors may elaborate the reaction mechanism as well as support their findings by stating that "The results are in compliance with the findings of Nasir of et al. [27] incorporating industrial waste-based pastes wherein adequate curing enabled dissolution of precursor materials which consequently led to formation of the dense skeletal matrix thereby yielding enhancement in overall engineering.

Response: Thanks for your comment. This section has been revised to:

“Fig. 11 shows the microstructure of different PG/RCA base materials. As observed, some small cracks appear in the microstructure of 0CPG-57RCA (Fig. 7a), larger cracks appear in 20PG-45RCA (Fig. 7b), and larger cracks still exist in 20CPG-45RCA (Fig. 7c), while the crack width of 20CPG-45RCA-11SMN is obviously smaller (Fig. 7d). The phenomena identified that the addition of SMN can promote the hydration process to assist the stabilization of PG/RCA roadbase material, by greatly improving the bonding properties between CPG and other components, for reducing the development of cracks. The results are in compliance with the findings of Nasir of et al. [37] incorporating industrial waste-based pastes wherein adequate curing enabled dissolution of precursor materials which consequently led to formation of the dense skeletal matrix thereby yielding enhancement in overall engineering. From the microstructural results, it is proved that using SMN to help recycle amounts of CPG into roadbase materials is technically feasible and worthy of being recommended.”

Round 2

Reviewer 1 Report

“Line 54: “Regarding the comprehensive utilization of PG and RCA in roadbase materials [16,17]”, what conclusions did these references obtain? How the study presented differ from these references? What is the novelty respect to those references?”

Have you included these references in the new version? Please, make easy to the reviewer to follow the answers. If it is necessary, explain the differences between old and new version. The expression “check it in the article” is not enough.

If the ref of the old manuscript [16,17] have been included, please, mention their finding s in the manuscript.

“Line 99: “which were composed of natural aggregates and adhered mortar” were not there any concrete particles? if that the case and there were just natural aggregates and mortar it is not regular RCA.”

I did mention that because of the great water absorption in the coarse fraction that it is in Table 3, depending on the amount of cement-based fragments and natural aggregates the properties of RCA can vary, it looks like the RCA that the authors have used has a great amount of cement-based fragments and low natural aggregates. It would be interesting to know their proportion.

Why the relation RCA: NFA=3:2 was in weight?

The material in studied is closer to a roller compacted concrete with a low cement content than to a cement stabilised since the aggregate are separated in two fractions (fine and coarse)

Line 164: “standard sample of 28d” was it dried or at the conditions of curing?” The issue is that such a porous materials like cement stabilized roadbase materials is strongly affected by the freeze-thaw cycles. Moreover, when it was introduced in these aggressive conditions in a saturated or at least, with a great moisture content, usually the samples are seriously damaged. Is there any reference in which similar samples were subjected to similar test method? If there is, which were the results?

Author Response

“Line 54: “Regarding the comprehensive utilization of PG and RCA in roadbase materials [16,17]”, what conclusions did these references obtain? How the study presented differ from these references? What is the novelty respect to those references?”

Have you included these references in the new version? Please, make easy to the reviewer to follow the answers. If it is necessary, explain the differences between old and new version. The expression “check it in the article” is not enough.

If the ref of the old manuscript [16,17] have been included, please, mention their findings in the manuscript.

Response: Thanks for your comments again. As reviewers suggested rewriting the introduction, the original [16,17] is not included in the new version. For the differences between old and new version, the introduction in the old version is now almost all updated in the new version including the deletion of [16,17] because of inappropriate citations.

“Line 99: “which were composed of natural aggregates and adhered mortar” were not there any concrete particles? if that the case and there were just natural aggregates and mortar it is not regular RCA.”

I did mention that because of the great water absorption in the coarse fraction that it is in Table 3, depending on the amount of cement-based fragments and natural aggregates the properties of RCA can vary, it looks like the RCA that the authors have used has a great amount of cement-based fragments and low natural aggregates. It would be interesting to know their proportion.

Response: Thanks for your comment again. For your concern the mass proportion of cement-based fragments to natural aggregates is approximately 25~30:100. The following changes have been done:

Table 3. Physical performance indicators of aggregates.

Aggregates

Bulk density (kg/m3)

Apparent density (kg/m3)

Porosity (%)

Water absorption rate (%)

Crushing value (%)

Mass percent of Adhered Mortar to NA (%)

Fineness modulus

RCA

1190

2560

0.47

6.1

15.9

25~30

NFA

1240

2597

2.94

Why the relation RCA: NFA=3:2 was in weight?

The material in studied is closer to a roller compacted concrete with a low cement content than to a cement stabilised since the aggregate are separated in two fractions (fine and coarse).

Response: Thanks for your comment again. We mentioned in this in our first response. This is because according to JTGT F20-2015, the composition of cement stabilized roadbase materials is 5% cement and 95% aggregates, where the coarse aggregate accounts for approximately 60% by weight of total aggregates for a typical aggregate gradation recommended in China. Therefore, our study aimed to replace all the coarse natural aggregates by coarse RCA particles. This is why the RCA content is 57% (95%×60%=57%).

If possible, you may check this Chinese standard entitled “Technical Guigelines for Construction of Highway RoadBases” (JTGT F20-2015).

If this answer is still not well satisfied with your questions, you may detail a little bit more of your comments to us for better understanding. Thanks very much.

Line 164: “standard sample of 28d” was it dried or at the conditions of curing?” The issue is that such a porous materials like cement stabilized roadbase materials is strongly affected by the freeze-thaw cycles. Moreover, when it was introduced in these aggressive conditions in a saturated or at least, with a great moisture content, usually the samples are seriously damaged. Is there any reference in which similar samples were subjected to similar test method? If there is, which were the results?

Response: Thanks for your comments. As mentioned in our resubmission, we had revised the descriptions to “According to JTG E51-2009, six samples were prepared in each group, and after curing for 28d, the cured samples were individually put into the -18°C cryogenic box for a 16h-freezing test. After this, the samples were taken out and immediately put it into a 20°C sink for melting for 8h.” For this concern we also added one citation [36] to the corresponding section “3.3. Frost resistance”, which presented the result similar to that obtained in our study.

To be detailed, Saberian et al. [36] found the unconfined compressive strengths (UCS) of samples were decreased after one freeze-thaw cycle (first frozen for 1d and then thawed for 1d), which after T-F cycle (first thawed for 1d and then refrozen for 1d) were significantly increased.

Reviewer 2 Report

Thank you for addressing the issues raised during the reviewing process. In my opinion, the manuscript fulfills the quality requirements for being published.

Author Response

Thank you for addressing the issues raised during the reviewing process. In my opinion, the manuscript fulfills the quality requirements for being published.

Response: Thank you for your affirmation.